# Skeleton interoception regulates bone and fat metabolism through hypothalamic neuroendocrine NPY

Xiao Lv[1,2†], Feng Gao[1,2†], Tuo Peter Li[1], Peng Xue[1], Xiao Wang[1], Mei Wan[1], Bo Hu[1], Hao Chen[1], Amit Jain[1], Zengwu Shao[2], Xu Cao[1]*

[1]Department of Orthopaedic Surgery, Institute of Cell Engineering, and Department of Biomedical Engineering, The Johns Hopkins University, Baltimore, United States; [2]Department of Orthopaedics, Union Hospital, Tongji Medical College, Huazhong University of Science and Technology, Wuhan, China

**Abstract** The central nervous system regulates activity of peripheral organs through interoception. In our previous study, we have demonstrated that PGE2/EP4 skeleton interception regulate bone homeostasis. Here, we show that ascending skeleton interoceptive signaling downregulates expression of hypothalamic neuropeptide Y (NPY) and induce lipolysis of adipose tissue for osteoblastic bone formation. Specifically, the ascending skeleton interoceptive signaling induces expression of small heterodimer partner-interacting leucine zipper protein (SMILE) in the hypothalamus. SMILE binds to pCREB as a transcriptional heterodimer on *Npy* promoters to inhibit NPY expression. Knockout of EP4 in sensory nerve increases expression of NPY causing bone catabolism and fat anabolism. Importantly, inhibition of NPY Y1 receptor (Y1R) accelerated oxidation of free fatty acids in osteoblasts and rescued bone loss in *Avil^Cre:Ptger4^{fl/fl}* mice. Thus, downregulation of hypothalamic NPY expression lipolyzes free fatty acids for anabolic bone formation through a neuroendocrine descending interoceptive regulation.

*For correspondence: xcao11@jhmi.edu

†These authors contributed equally to this work

Competing interests: The authors declare that no competing interests exist.

## Introduction

The central nervous system (CNS) regulates activity of peripheral organs to control the internal state of the body has been referred to interoception as an emerging science (*Chen et al., 2021*). CNS receives interoceptive signals originating from peripheral tissues by ascending pathways for interpretation and integration to regulate the activity of specific organs through descending pathways (*Chen et al., 2021*). The skeleton is the largest organ in the body and stores vital minerals, forms muscle attachments and comprises the niches for hematopoiesis (*Leider, 1947*). Therefore, CNS regulation of skeleton homeostasis is particularly critical in coordination of activities of the endocrine system and functional related organs. The skeleton is innervated abundantly by sensory nerves, not only to sense pain, but to contribute to bone anabolism (*Brazill et al., 2019*; *Fukuda et al., 2013*). The loss of sensory nerves in bone impairs bone mass formation (*Fukuda et al., 2013*). In our recent studies, we have established that sensory nerves in the bone perceive the concentration changes of osteoblast-derived prostaglandin E2 (PGE2) to activate PGE2 receptor EP4 as ascending interoceptive signal to the hypothalamus. It has been shown that inhibition of dopamine hydroxylase (DBH) which the inability to synthesize catecholamines in the hypothalamus tunes down the sympathetic activity to induce osteoblastic bone formation (*Takeda et al., 2002*). We have demonstrated that PGE2/EP4 ascending interoceptive signal downregulates sympathetic tone for osteoblastic bone formation to maintain bone homeostasis as descending interoceptive signal (*Chen et al., 2019*; *Hu et al., 2020*). Importantly, this descending interoceptive signal regulates lineage commitment of mesenchymal stem/stromal cells (MSCs) between osteoblasts and

adipocytes (*Hu et al., 2020*). The PGE2/EP4 skeleton interoception could be an essential CNS regulation of bone homeostasis.

Bone is an endocrine organ, and its metabolism is regulated endocrine system. It is imperative to know whether the skeleton interoception directly regulates hypothalamic endocrine activity. Particularly, osteoblastic bone formation is an energy-consuming process involved in metabolic activity of different organs and tissues including kidney, liver, and fat etc, particularly requiring 20% fatty acid oxidation of total energy consumption (*Adamek et al., 1987*; *Kim et al., 2017*). Bone is constantly under remodeling, which is an energy-consuming process that maintains bone and calcium metabolic homeostasis through activities such as bone matrix synthesis, mineralization, and osteoclastic bone resorption. Disruption in skeletal energetic balance induces metabolic diseases (*Karner and Long, 2018*; *Riddle and Clemens, 2017*; *Confavreux et al., 2009*; *Lee et al., 2007*; *Zaidi, 2007*). The increasing prevalence of diabetes, obesity and other metabolic disorders has focused attention on energy metabolism and whole-organism bioenergetic homeostasis (*Hu et al., 2003*). During the past 15 years, endocrine hormones, such as insulin and parathyroid hormone, have been recognized as master regulators of energy metabolism (*Ferron et al., 2010*; *Guilherme et al., 2019*). Likewise, specific hypothalamic neurons associated with appetite and pleasure communicate information via the autonomic nervous system to coordinate energy transport between major energy centers, including the liver, heart, fat tissue, and skeleton (*Yadav et al., 2009*; *Ducy et al., 2000*; *Guilherme et al., 2019*). Leptin-deficient mice have altered bone mass because of dysregulation of the hypothalamic-sympathetic-bone axis (*Ducy et al., 2000*; *Takeda et al., 2002*). Although many signaling and metabolic mechanisms in specific organs have been determined, how the activities in different tissues are coordinated such as NPY signaling regulation of bone mass with promotion of lipolysis in fat tissue is still largely unknown.

The hypothalamus controls whole-body energy homeostasis by integrating peripheral information and coordinating energy transfer among organ centers. Neuropeptide Y (NPY) is one of the most abundant neuropeptides in the hypothalamus and is widely expressed in the central and peripheral nervous systems (*Ekblad et al., 1984*). NPY concentration is highest in the neurons of the arcuate nucleus (ARC) in the hypothalamus, which functions as an orexigenic peptide that induces food intake (*Zhang et al., 2019*). The role of NPY in regulating whole-body energy metabolism through the CNS has been studied extensively (*Loh et al., 2015*; *Riediger, 2012*). NPY promotes energy storage in white adipose tissue; knockout of NPY in the hypothalamus promotes thermogenesis and energy expenditure and prevents obesity (*Zhang et al., 2014*; *Chao et al., 2011*; *Park et al., 2014*). Knockout of the NPY Y1 receptor (Y1R) in adipocytes showed resistance to diet-induced obesity, whereas activation of Y1R stimulates fat accretion (*Zhang et al., 2014*). Y1R antagonist BIBO3304 enhanced energy expenditure and improves glucose homeostasis (*Yan et al., 2021*). Conversely, bone mass increased significantly in NPY knockout mice (*Baldock et al., 2009*). Moreover, bone mass increased significantly in NPY Y1R knockout mice and wild-type (WT) mice treated with Y1R inhibitor because Y1R is expressed in the osteoblasts (*Brothers and Wahlestedt, 2010*; *Sousa et al., 2020*). Importantly, embryonic stem cells accelerated adipogenesis induced by NPY system activation, which is consistent with our previous observation (*Han et al., 2012*; *Hu et al., 2020*). Given that reduction of NPY concentration in the hypothalamus induces catabolism of adipose tissue and osteoblastic bone formation, NPY could control the balance between osteoblastic bone formation and fat metabolism through skeletal interoception.

In the current study, we found that NPY expression in the hypothalamus was regulated by PGE2/EP4 ascending interoceptive signaling to balance bone and fat metabolism. Downregulation of hypothalamic NPY expression induces lipolysis of white adipose tissue in promoting osteoblast fatty acid (FA) uptake and bone formation. Moreover, Inhibition of NPY receptor Y1R accelerates osteogenesis and mineralization. Thus, downregulation of hypothalamic NPY expression by ascending skeleton interoceptive signaling induces adipose tissue lipolysis for osteoblastic bone formation as descending neuroendocrine interoceptive pathway.

## Results

### Sensory nerve denervation induces NPY expression in the hypothalamus

To determine the skeletal interoception from femur to the hypothalamus, an anterograde multisynaptic tracer herpes simplex virus type 1 (HSV-1) H129-G4 was directly injected in the femur marrow in 3-month-old wild-type mice (*Zeng et al., 2017*), GFP labeled neurons in ARC area has been detected at 5 days post infection (dpi) (*Figure 1A*). This result confirmed skeletal interoception between femur and the hypothalamus. To investigate potential sensory regulation of hypothalamus derived neuropeptides, we crossed nerve growth factor receptor TrkA floxed (*Ntrk1$^{fl/fl}$*) mice with sensory neuron-specific Cre mice (Advilin$^{Cre}$, *Avil$^{Cre}$*) to generate sensory denervation mice (*Avil$^{Cre}$: Ntrk1$^{fl/fl}$*) (*Chen et al., 2019*). Knockout efficiency was confirmed by immunofluorescence staining of TrkA in dorsal root ganglia neurons isolated from *Avil$^{Cre}$:Ntrk1$^{fl/fl}$* mice (*Figure 1—figure supplement 1A*). Serum NPY concentration was significantly elevated in 3-month-old *Avil$^{Cre}$:Ntrk1$^{fl/fl}$* mice compared with that of WT littermates (*Figure 1B*). Whereas serum NuM, CRH and CART concentration did not show significantly difference between 3-month-old *Avil$^{Cre}$:Ntrk1$^{fl/fl}$* and *Ntrk1$^{fl/fl}$* (*Figure 1—figure supplement 1B*). Immunostaining of the hypothalamus showed that NPY expression was also significantly increased in the ARC of *Avil$^{Cre}$:Ntrk1$^{fl/fl}$* mice (*Figure 1C*). Moreover, food intake increased in *Avil$^{Cre}$:Ntrk1$^{fl/fl}$* mice compared with WT littermates (*Figure 1D*). Whereas total body weight remained unchanged (*Figure 1E*), the size and weight of gonadal and inguinal fat pads increased significantly in *Avil$^{Cre}$:Ntrk1$^{fl/fl}$* mice compared with WT mice (*Figure 1F*). Accordingly, echo magnetic resonance imaging (qNMR) detected a significant increase in the fat mass of adult (3-month-old) but not young (1-month-old) *Avil$^{Cre}$:Ntrk1$^{fl/fl}$* mice (*Figure 1G*). Interestingly, no significant differences in lean mass were found in the 1- or 3-month-old groups (*Figure 1H*).

To confirm sensory nerve regulation of hypothalamic NPY expression in adult mice, we crossed Advilin$^{Cre}$ (*Avil$^{Cre}$*) mice with *Rosa26$^{lsl-DTR}$* mice to generate inducible sensory denervation mice (*Avil$^{Cre}$: Rosa26$^{lsl-DTR}$*). Sensory denervation was induced in adult *Avil$^{Cre}$: Rosa26$^{lsl-DTR}$* mice by injection of diphtheria toxin (DTX) (*Chen et al., 2019*). Indeed, NPY levels in serum increased significantly 5 and 7 days after DTX injection (*Figure 1I*), and NPY expression in the ARC increased significantly 7 days after DTX injection, as shown by immunostaining of hypothalamus sections (*Figure 1J*). Daily food intake also increased significantly in the DTX group (*Figure 1K*). Consistent with results in *Avil$^{Cre}$:Ntrk1$^{fl/fl}$* mice, gonadal and inguinal fat pad size and weight increased significantly (*Figure 1L*), and qNMR validated much higher fat mass in the DTX group (*Figure 1M*), whereas no significant difference in lean mass was found between the DTX and vehicle groups (*Figure 1N*). Our results show that peripheral sensory nerves regulate NPY expression in the hypothalamus ARC for metabolism of adipose tissue.

### Deletion of EP4 receptor in the sensory nerve increase NPY expression in the hypothalamus ARC

Activation of EP4 receptor in sensory nerves by PGE2 maintain the balance of osteogenesis and adipogenesis (*Chen et al., 2019*; *Hu et al., 2020*). PGE2 concentration elevated with injection of SW033291 as it inhibits 15-hydroxyprostaglandin dehydrogenase for degradation of PGE2 (*Zhang et al., 2015*; *Blackwell et al., 2010*; *Uppal et al., 2008*). To confirm that SW033291 elevated PGE2 concentration in bone marrow rather than in the hypothalamus, we measured PGE2 concentrations in bone marrow and the hypothalamus after treatment for 3 hr. The SW033291 group had a significantly higher PGE2 concentration in bone marrow compared with the vehicle group (*Figure 2A*, left), but no difference in PGE2 concentration was found between groups when analyzing the hypothalamic interstitium (*Figure 2A*, right). Compared with the vehicle group, WT mice treated with SW033291 had significantly lower *Npy* gene expression in the ARC and significantly lower serum NPY level (*Figure 2B and C*).

To examine whether NPY expression in the hypothalamus is regulated by skeletal interoception, we generated sensory nerve EP4 knockout mice (*Avil$^{Cre}$:Ptger4$^{fl/fl}$*) by crossing *Ptger4$^{fl/fl}$* mice (*Ptger4* is the gene that encodes EP4 receptor) with *Avil$^{Cre}$* mice. Serum NPY concentration and its expression in the ARC area were significantly higher in *Avil$^{Cre}$:Ptger4$^{fl/fl}$* 3-month-old mice compared with that of their WT littermates *Ptger4$^{fl/fl}$*, but no significant differences were found between the

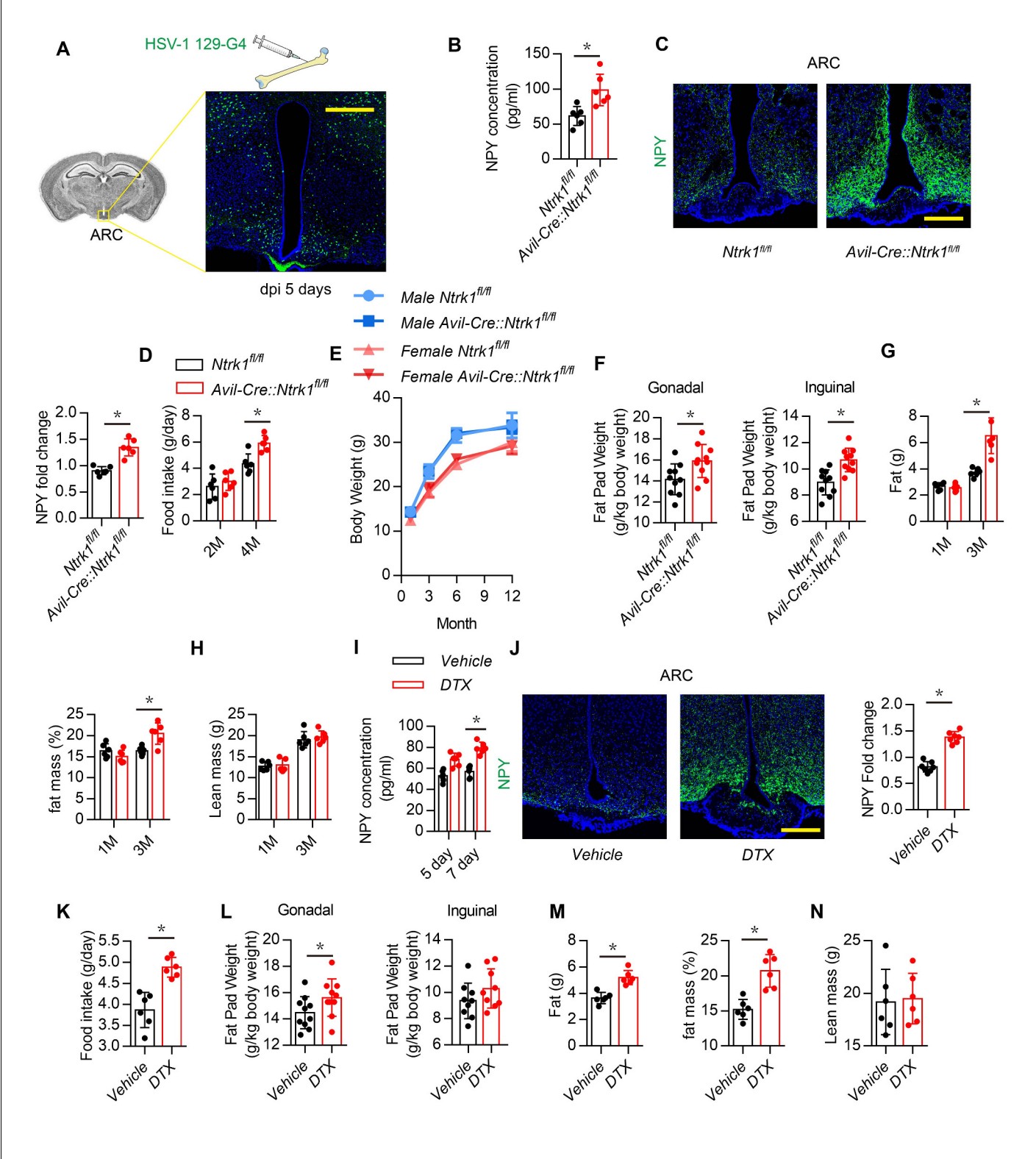

**Figure 1.** Sensory nerve denervation induces NPY expression. (**A**) Representative images of GFP+ neurons in the hypothalamus after multisynaptic tracer HSV-1 H19-G4 injected in the femur marrow for 5 days (dpi, days post injection). (**B**) Enzyme-linked immunosorbent assay (ELISA) analysis of NPY level in serum in 3-month-old *Ntrk1*$^{fl/fl}$ and *Avil*$^{Cre}$:*Ntrk1*$^{fl/fl}$ mice. (**C**) Representative images of immunofluorescence staining and quantitative analysis of NPY (*green*) in the ARC of hypothalamus of 3-month-old *Ntrk1*$^{fl/fl}$ and *Avil*$^{Cre}$:*Ntrk1*$^{fl/fl}$ mice. DAPI stains nuclei blue. Scale bars = 50 μm. (**D**) Quantitative

*Figure 1 continued on next page*

*Figure 1 continued*

analysis of food intake for 2- and 4-month-old *Ntrk1*[fl/fl] and *Avil*[Cre]*:Ntrk1*[fl/fl] mice. (E) Quantitative analysis of body weight for male and female *Ntrk1*[fl/fl] and *Avil*[Cre]*:Ntrk1*[fl/fl] mice at 1, 3, 6, and 12 months old. (F) Quantitative analysis of the weight of the gonadal and inguinal fat pads isolated from 3-month-old *Ntrk1*[fl/fl] and *Avil*[Cre]*:Ntrk1*[fl/fl] mice. qNMR analysis of (G) fat weight, fat mass, and (H) lean mass of 1- and 3-month-old *Ntrk1*[fl/fl] and *Avil*[Cre]*: Ntrk1*[fl/fl] mice. (I) ELISA analysis of serum NPY level of 3-month-old *Avil*[Cre]*:Rosa26* [lsl-DTR] mice injected with vehicle or 1 ug/kg/d of DTX for 5 and 7 days. (J) Representative images of immunofluorescence staining and quantitative analysis of NPY (*green*) in the ARC of hypothalamus of 3-month-old *Avil*[Cre]*: Rosa26* [lsl-DTR] mice injected with vehicle or 1 µg/kg/d DTX for 7 days. Scale bars = 50 µm. (K) Quantitative analysis of food intake for 3-month-old *Avil*[Cre]*:Rosa26* [lsl-DTR] mice injected with vehicle or DTX for 1 month. (L) Quantitative analysis of the weight of the gonadal and inguinal fat pads isolated from 3-month-old *Avil*[Cre]*:Rosa26* [lsl-DTR] mice injected with vehicle or DTX for 1 month. qNMR analysis of (M) fat weight, fat mass, and (N) lean mass of 3-month-old *Avil*[Cre]*:Rosa26* [lsl-DTR] mice injected with vehicle or DTX for 1 month. N $\geq$ six per group. *p < 0.05, and N.S. means not significant. Statistical significance was determined by Student's t-test.

The online version of this article includes the following source data and figure supplement(s) for figure 1:

**Source data 1.** Raw data of neuropeptiedes level.

**Figure supplement 1.** Knockout efficiency of *Avil*[Cre]*:Ntrk1*[fl/fl] mice.

altered 1-month-old mice and the WT 1-month-old mice (*Figure 2D*). To assess whether elevated local PGE2 activates EP4 signaling in sensory nerves to regulate NPY in the CNS, we administered SW033291. The effect of PGE2 on decreased NPY expression was abolished in *Avil*[Cre]*:Ptger4*[fl/fl] mice (*Figure 2E and F*). Food intake significantly higher in *Avil*[Cre]*:Ptger4*[fl/fl] compared with that of their WT littermates, but no significant change was found in *Avil*[Cre]*:Ptger4*[fl/fl] mice after SW033291 injection (*Figure 2G*). Moreover, no significant change in total body weight was found in male and female mice in both *Avil*[Cre]*:Ptger4*[fl/fl] and their WT littermates (*Figure 2H*). Gonadal and inguinal fat pad weights increased significantly in *Avil*[Cre]*:Ptger4*[fl/fl] mice, whereas injection of SW033291 reduced major fat pad weights in WT mice but not in *Avil*[Cre]*:Ptger4*[fl/fl] mice (*Figure 2I*). Again, qNMR showed that fat mass increased significantly in *Avil*[Cre]*:Ptger4*[fl/fl] mice, and injection of SW033291 reduced fat mass in WT mice but not in the *Avil*[Cre]*:Ptger4*[fl/fl] mice (*Figure 2J*). Similarly, no significant differences in lean mass were found between WT and *Avil*[Cre]*:Ptger4*[fl/fl] mice with or without SW033291 injection (*Figure 2K*). We have shown that an increase in PGE2 caused by injection of SW033291 activates skeletal interoception in *Ptger4*[fl/fl] mice but not in *Avil*[Cre]*:Ptger4*[fl/fl]. Thus, NPY expression in the ARC was regulated by PGE2/EP4 ascending interoceptive signaling in balance bone and adipose tissue metabolism.

## PGE2/EP4 ascending interoceptive signaling induces expression of transcriptional repressor SMILE

To examine the mechanism of PGE2 in bone marrow in regulating NPY in the ARC, we analyzed the transcriptional mechanism of NPY by stimulating skeletal interoception by PGE2. PGE2 in the bone stimulates hypothalamic CREB phosphorylation. SMILE is a transcription corepressor forming the heterodimer with phosphorylated cyclic AMP–response element binding protein (pCREB) in suppression of gene transcription (*Lee et al., 2018*; *Misra et al., 2011*). To determine whether peripheral administration of SW033291 increased the expression of SMILE in the hypothalamus, we measured hypothalamic CREB phosphorylation and SMILE protein expression in mice injected with SW033291 through western blot analysis. CREB phosphorylation and SMILE protein expression increased significantly in mice injected with SW033291 (*Figure 3A*). To investigate the transcriptional mechanism, we performed the chromatin immunoprecipitation (ChIP) assay with three potential pCREB-binding sites (*Table 1*: primers 1–3) in the *Npy* gene promoter (*Figure 3B*). ChIP assay results showed that SW033291 induced specific binding of pCREB/SMILE to the most distal CREB binding site of the *Npy* promoter (*Figure 3C–E*). Finally, we analyzed the levels of pCREB and SMILE and expression of NPY in the ARC of *Avil*[Cre]*:Ptger4*[fl/fl] and *Ptger4*[fl/fl] mice peripherally injected with vehicle or SW033291. Immunostaining of hypothalamus sections showed that expression of both SMILE and CREB phosphorylation were increased significantly in the ARC of *Ptger4*[fl/fl] mice but not in *Avil*[Cre]*: Ptger4*[fl/fl] mice (*Figure 3F–G*). Taken together, these results indicate that PGE2/EP4 ascending skeleton interoceptive signaling induces transcriptional repressor SMILE expression in the hypothalamus, which interacts with pCREB as a heterodimer to bind to *Npy* promoter therefore repressing *Npy* gene transcription.

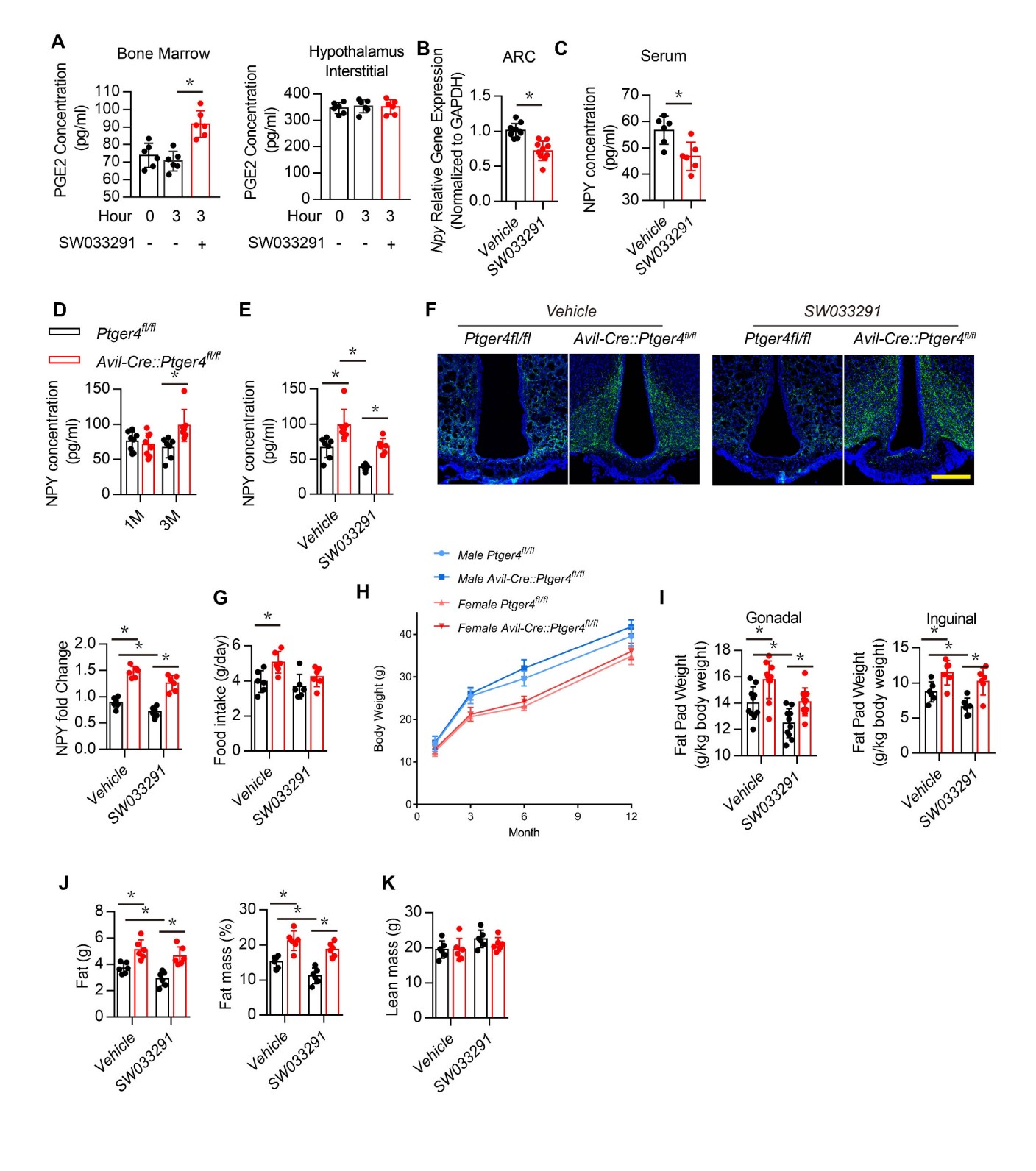

**Figure 2.** Deletion of EP4 receptor in sensory nerve increases NPY expression. (**A**) ELISA analysis of bone marrow and hypothalamus interstitial in WT mice treated with vehicle or SW033291 for 0 and 3 hr. (**B**) RT-PCR quantitative analysis of *Npy* gene expression in the ARC area in 3-month-old male WT mice after being treated with vehicle or 10 mg/kg/d SW033291 for 1 month. (**C**) ELISA analysis of NPY level in serum from 3-month-old male WT mice after being treated with vehicle or 10 mg/kg/day SW033291 for 1 month. (**D**) ELISA analysis of NPY level in serum from 1- and 3-month-old *Ptger4*[fl/fl]

*Figure 2 continued on next page*

*Figure 2 continued*

and *Avil^Cre^:Ptger4^fl/fl^* mice. (E) ELISA analysis of NPY level in serum from 3-month-old *Ptger4^fl/fl^* and *Avil^Cre^:Ptger4^fl/fl^* mice treated with vehicle or 10 mg/kg/d SW033291 for 1 month. (F) Representative images of immunofluorescence staining and quantitative analysis of NPY (*green*) in the ARC of hypothalamus of 3-month-old *Ptger4^fl/fl^* and *Avil^Cre^:Ptger4^fl/fl^* mice treated with vehicle or 10 mg/kg/d SW033291 for 1 month. DAPI stains nuclei blue. Scale bars = 50 µm. (G) Quantitative analysis of food intake for 3-month-old *Ptger4^fl/fl^* and *Avil^Cre^:Ptger4^fl/fl^* mice treated with vehicle or 10 mg/kg/d SW033291 for 1 month. (H) Quantitative analysis of body weight for male and female *Ptger4^fl/fl^* and *Avil^Cre^:Ptger4^fl/fl^* mice at 1, 3, 6, and 12 months old. Quantitative analysis of the weight of the (I) gonadal and inguinal fat pads isolated from 3-month-old *Ptger4^fl/fl^* and *Avil^Cre^:Ptger4^fl/fl^* mice treated with vehicle or SW033291 for 1 month. qNMR analysis of (J) fat weight, fat mass, and (K) lean mass of 3-month-old *Ptger4^fl/fl^* and *Avil^Cre^:Ptger4^fl/fl^* mice treated with vehicle or SW033291 for 1 month. N ≥ six per group. *$p < 0.05$, and N.S. means not significant. Statistical significance was determined by Student's t-test for A–D. Statistical significance was determined by two-way analysis of variance for E-G, I-K.

The online version of this article includes the following source data and figure supplement(s) for figure 2:

**Source data 1.** Raw data of quantification of NPY level, staining of NPY, perilipin, osteocalcin and pCREB, Fatp1 and cpt1b gene expression.
**Figure supplement 1.** Osteocytes derived PGE2 did not affects hypothalamic NPY.

## Downregulation of hypothalamic NPY by ascending skeleton interoception promotes osteoblastic bone formation

NPY induces catabolic activity in fat tissue (*Zhang et al., 2014*). Therefore, we investigated whether hypothalamic NPY regulated by PGE2 coordinates metabolism between bone and fat. Injection of SW033291 significantly increased bone volume and trabecular bone number in WT mice but not in *Avil^Cre^:Ptger4^fl/fl^* mice (*Figure 3—figure supplement 1A–B*). Calcein double-labeling confirmed that SW033291-induced bone formation and mineral composition in WT mice was abolished in *Avil^Cre^: Ptger4^fl/fl^* mice (*Figure 4A and B*). Co-immunostaining of trabecular bone sections showed that pCREB in osteocalcin^+^ osteoblastic cells was significantly increased in WT mice injected with SW033291, and No such increase in pCREB in osteoblastic cells occurred in *Avil^Cre^:Ptger4^fl/fl^* mice (*Figure 4C and D*). Decrease of phosphorylation of CREB in osteoblastic cells was also confirmed in *Avil^Cre^:Ptger4^fl/fl^* mice relative to WT littermates. SW033291-induced pCREB level was abolished in *Avil^Cre^:Ptger4^fl/fl^* mice (*Figure 3—figure supplement 1C*). Similarly, cAMP production was stimulated by SW033291 in bone marrow, and the stimulation was abolished in *Avil^Cre^:Ptger4^fl/fl^* mice (*Figure 4E*). Reverse transcription-polymerase chain reaction (RT-PCR) showed that the expression of osteogenic markers *Runx2* (*Figure 4F*), alkaline phosphatase (*Alp*), and Collagen type Ia (*Col1a1*) (*Figure 3—figure supplement 1D*) increased significantly in WT mice with SW033291 injection, and these effects were eliminated in *Avil^Cre^:Ptger4^fl/fl^* mice. Importantly, co-immunostaining of perilipin and osteocalcin showed that the effects of SW033291 on osteogenesis and adipogenesis inhibition were significantly abrogated in *Avil^Cre^:Ptger4^fl/fl^* mice (*Figure 4G and H*). We also studied FA uptake in osteoblasts. FA-oxidation-associated gene *Fatp1*, which facilitates long-chain FA uptake into cells, and *Cpt1b*, which helps FA transport on the mitochondrial membrane, were measured in osteoblasts (*Stahl et al., 2001*). As expected, SW033291 injection in WT mice caused significantly increased *Fatp1* and *Cpt1b* expression in osteoblasts, but these effects were not present in *Avil^Cre^:Ptger4^fl/fl^* mice (*Figure 4I*). These data show that SW033291 induced bone formation accompanied by FA uptake in osteoblasts. Taken together, our results indicate that NPY downregulated by PGE2/EP4 ascending interoceptive signaling regulates osteoblastic energy homeostasis to induce bone formation.

## Downregulation of NPY by ascending skeleton interoceptive signal stimulates lipolysis of adipose tissue

Skeletal interoception has been shown to balance the differentiation of MSCs between osteoblasts and adipocytes (*Hu et al., 2020*). We therefore investigated whether downregulation of NPY in the ARC by skeletal interoception induces lipolysis and osteoblastic bone formation. Expression of *Lipe* (hormone sensitive lipase, hsl), which hydrolyses intracellular triglycerides into free FAs, and *Pnpla2* (adipose triglyceride lipase, atgl), increased significantly with injection of SW033291 compared with that of vehicle mice, and the lipolysis activity was abrogated in the *Avil^Cre^:Ptger4^fl/fl^* mice (*Figure 5A and B*). FA concentration in circulation was higher in *Avil^Cre^:Ptger4^fl/fl^* mice compared with their WT littermates (*Figure 5C*). FA concentration decreased significantly with injection of SW033291 in WT mice but not in *Avil^Cre^:Ptger4^fl/fl^* mice (*Figure 5C*), suggesting that lipolysis of adipose tissue facilitates osteoblastic differentiation.

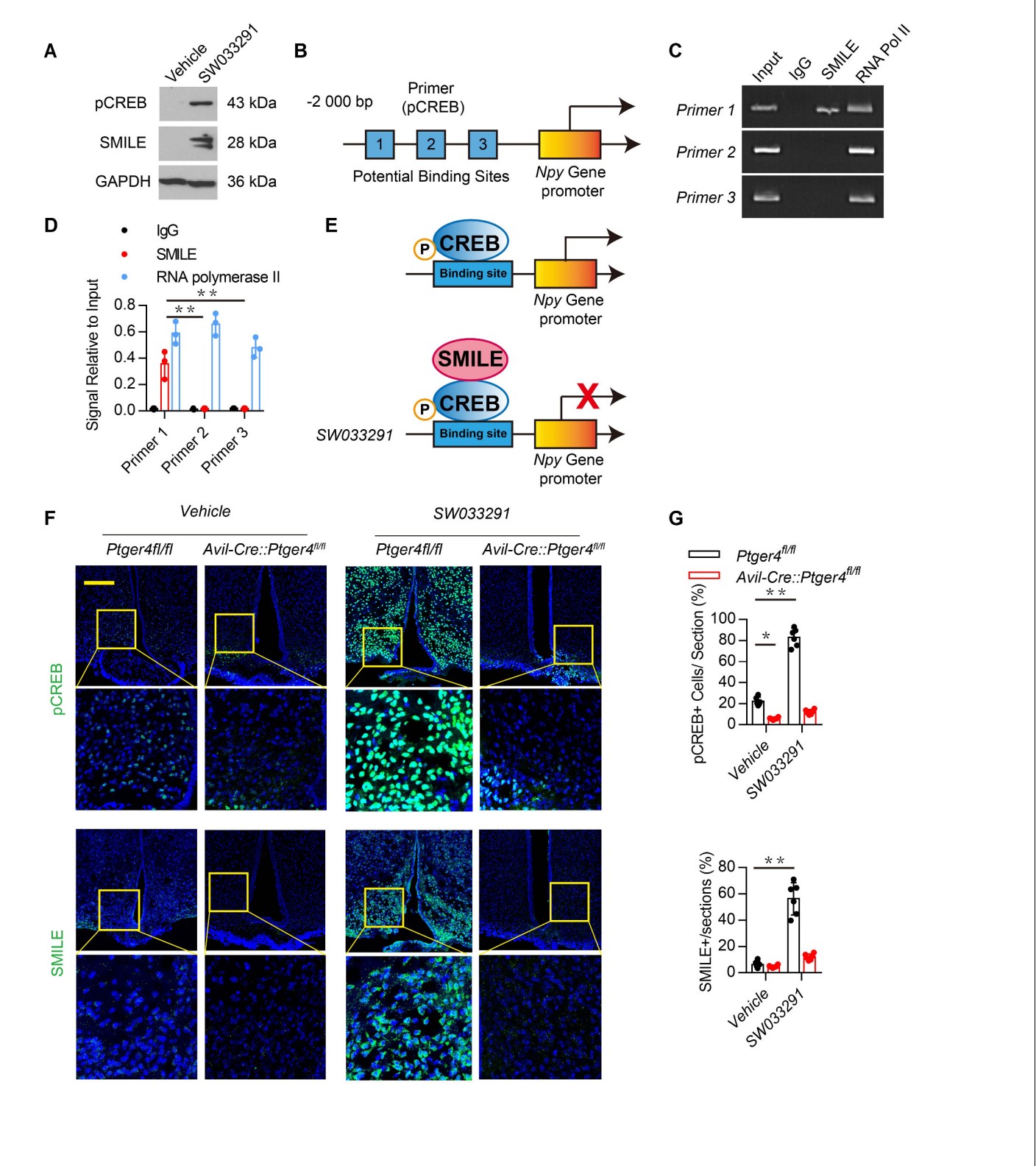

**Figure 3.** Stimulation of skeletal interoception induces SMILE to suppress NPY transcription in the hypothalamus. (**A**) Western blot analysis expression of phosphorylation CREB and SMILE in the ARC area from 3-month-old male WT mice injected with vehicle or 10 mg/kg/day SW033291 for 3 days. (**B**) Diagram of potential pCREB binding site on the *Npy* gene promoter. (**C, D**) ChIP and RT-PCR quantitative analysis of pCREB on *Npy* gene promoter in the ARC area treated with 10 mg/kg/day SW033291 for 3 days. (**E**) Diagram of the mechanism of SW033291 down-regulated *Npy* gene expression in

*Figure 3 continued on next page*

*Figure 3 continued*

the ARC area. (F) Representative images of immunofluorescence staining and (G) quantitative analysis of the pCREB (up) and SMILE (down) in the ARC of the hypothalamus of 3-month-old *Ptger4^fl/fl^* and *Avil^Cre^:Ptger4^fl/fl^* mice treated with vehicle or 10 mg/kg/day SW033291 for 7 days. Scale bars = 50 μm. N ≥ six per group. *p < 0.05, and N.S. indicates not significant. Statistical significance was determined by two-way analysis of variance for D,G. The online version of this article includes the following source data and figure supplement(s) for figure 3:

**Source data 1.** Raw data of quantification of BV/TV and trabecular number and osteogeneic related gene expression.
**Figure supplement 1.** PGE2 regulate osteoblastic bone formation via EP4 receptor on sensory nerves.

Energy catabolic marker phosphorylation AMP-activated protein kinase (pAMPK) was further analyzed in marrow fat cells and white adipose tissues (*Daval et al., 2006*). Immunostaining of white adipose tissue sections showed that pAMPK expression was significantly increased in the perilipin⁺ adipocytes in WT mice, but the catabolic effect of SW033291 on adipocytes was impaired in *Avil^Cre^: Ptger4^fl/fl^* mice (*Figure 5D and E*). To validate deposition of FAs in adipose tissue in the bone marrow, we performed osmium tetroxide (OsO4) staining of fat droplets in decalcified femurs captured by μCT, which showed that bone marrow fat decreased significantly with injection of SW033291 in WT mice, but this effect was abrogated in *Avil^Cre^:Ptger4^fl/fl^* mice (*Figure 5F and G*). Collectively, these data indicate that stimulation of skeletal interoception induces adipose catabolic activity in bone marrow and white adipose tissue by downregulating NPY expression in the hypothalamus.

We have previously shown that osteoblasts derived PGE2 primarily involved in sensory nerve regulate osteoblasts differentiation (*Chen et al., 2019*; *Hu et al., 2020*). To determine whether PGE2 derived from osteoblasts regulates NPY expression in hypothalamus promotes osteoblastic energy homeostasis and fatty acid oxidation, *Bglap^Cre^:Cox2^fl/fl^* was generated to specifically knock out COX2 in osteoblastic cells. We found NPY expression significantly increased in serum and ARC in *Bglap^Cre^:Cox2^fl/fl^* relative to the WT group (*Figure 5H and I*). Immunofluorescence also demonstrated decrease of osteoblastic differentiation along with adipocytes accumulation in *Bglap^Cre^: Cox2^fl/fl^* as evidenced by decrease of osteocalcin positive osteoblasts and increase of perilipin positive adipocytes (*Figure 5J*). Moreover, expression of FA oxidation related gene *Cpt1b* and *Fatp1* in osteoblasts significantly decreased in *Bglap^Cre^:Cox2^fl/fl^* relative to WT group (*Figure 5K*). Taken together, these results showed that osteoblasts derived PGE2 regulate hypothalamic NPY expression via skeleton sensory interoceptive signals maintain osteoblastic energy homeostasis and FA oxidation.

## Inhibition of NPY Y1R accelerated oxidation of free FAs and rescued bone loss in *Avil^Cre^:Ptger4^fl/fl^* mice

Inhibition of Y1R promotes osteoblastic bone formation (*Sousa et al., 2012*; *Xie et al., 2020*). Therefore, we examined whether downregulation of NPY by skeletal interoception increases bone formation by reducing Y1R downstream signaling. High affinity NPY Y1R inhibitor (BIBO3304) was injected in *Avil^Cre^:Ptger4^fl/fl^* mice daily for 1 month. Interestingly, the bone loss phenotype in *Avil^Cre^: Ptger4^fl/fl^* mice was rescued by BIBO3304, as shown by μCT (*Figure 6A and B*), whereas elevation of PGE2 by injection of SW033291 did not rescue the bone loss in *Avil^Cre^:Ptger4^fl/fl^* mice (*Figure 4A* through 4D). This observation suggests that the mechanism of NPY-induced osteoblastic bone formation is distinct from that of PGE2/EP4 interoception. We then investigated whether the number of osteoblasts increased in response to stimulation of commitment of MSCs to osteoblast lineage cells by PGE2/EP4 interoception (*Hu et al., 2020*). Immunohistologic analysis showed that the number of osteocalcin⁺ osteoblasts did not change with BIBO3304 in *Avil^Cre^:Ptger4^fl/fl^* mice (*Figure 6C and D*), suggesting that NPY likely promotes osteoblast differentiation.

**Table 1.** Potential binding-sites for pCREB on *Npy* promoter.

| Primer | Forward | Reverse |
|---|---|---|
| 1 | AGGATCGCATATTGAAACA | ACTAACTCTGCAAGGGCAT |
| 2 | GAATCTTTCAAACATCCGA | TCCTGAAATCATTGGTAGC |
| 3 | GCTAAATCCAGGCTTCAACT | CCAGAACAACAATATCCCTC |

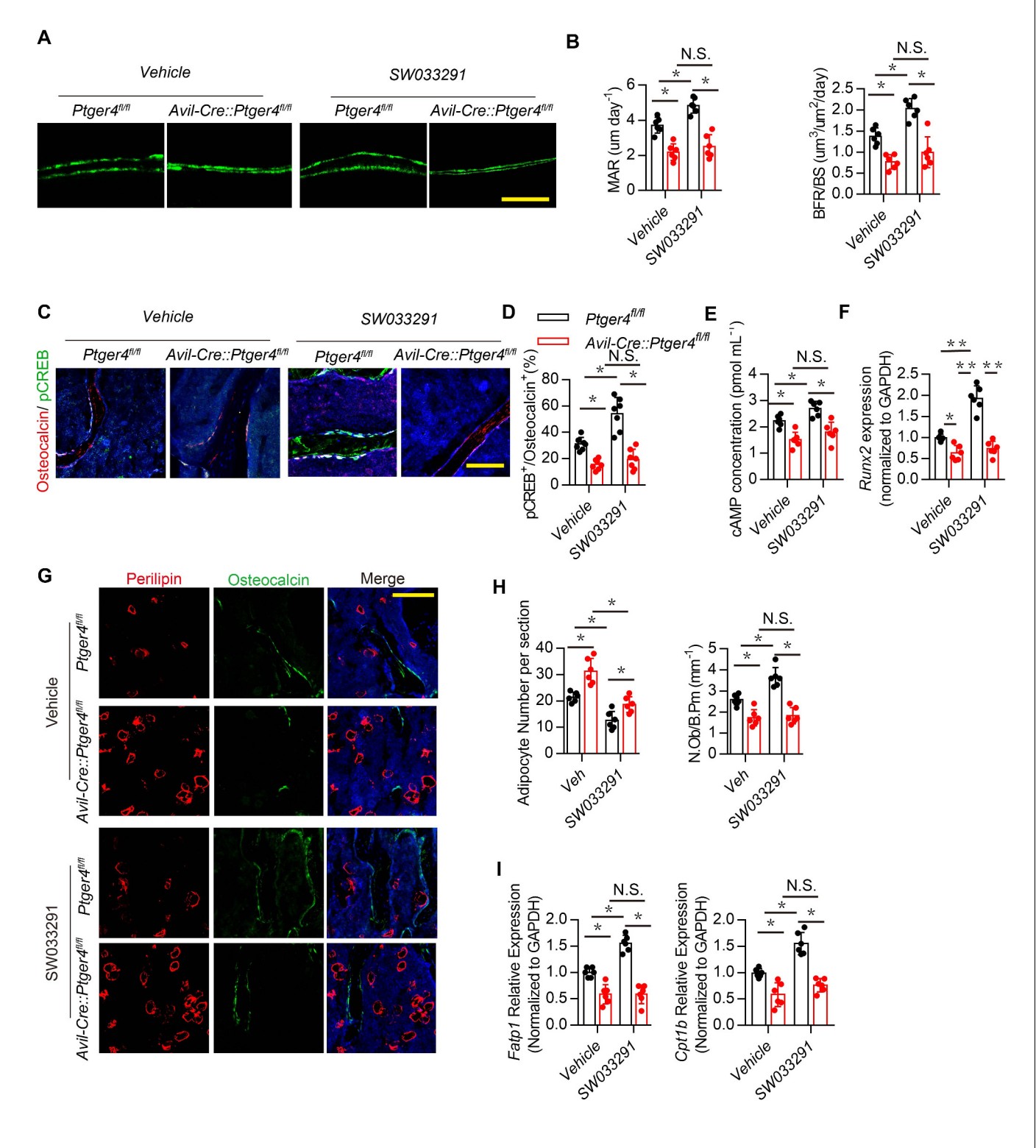

**Figure 4.** Skeletal interoception promotes osteoblastic bone formation by downregulation of hypothalamic NPY. (**A**) Representative images of calcein double labeling of femoral trabecular bone and (**B**) quantification of mineral apposition rate (MAR) and bone formation rate per bone surface (BFR/BS) in 3-month-old male *Ptger4^{fl/fl}* and *Avil^{Cre}:Ptger4^{fl/fl}* mice injected with vehicle or 10 mg/kg/d SW033291 for 1 month. Scale bars = 20 μm. (**C**) Representative co-immunofluorescence staining and (**D**) quantitative analysis of pCREB (*green*) and osteocalcin (*red*) from femurs of 3-month-old male *Ptger4^{fl/fl}* and *Avil^{Cre}:Ptger4^{fl/fl}* mice injected with vehicle or 10 mg/kg/day SW033291 for 1 month. Scale bars = 50 μm. (**E**) Quantitative analysis of cAMP

*Figure 4 continued on next page*

Figure 4 continued

concentration in the osteoblasts isolated from 3-month-old male *Ptger4*[fl/fl] and *Avil*[Cre]:*Ptger4*[fl/fl] mice injected with vehicle or 10 mg/kg/day SW033291 for 1 month. (F) RT-PCR quantitative analysis of Runx2 expression in femurs from 3-month-old male *Ptger4*[fl/fl] and *Avil*[Cre]:*Ptger4*[fl/fl] mice injected with vehicle or 10 mg/kg/day SW033291 for 1 month. (G) Representative co-immunofluorescence staining and (H) quantitative analysis (adipocyte number per section and number of osteoblast) of osteocalcin (*green*) and perilipin (*red*) from femurs of 3-month-old male *Ptger4*[fl/fl] and *Avil*[Cre]:*Ptger4*[fl/fl] mice injected with vehicle or 10 mg/kg/day SW033291 for 1 month. Scale bars = 50 µm. (I) RT-PCR quantitative analysis of *Fatp1* and *Cpt1b* expression in femurs from 3-month-old male *Ptger4*[fl/fl] and *Avil*[Cre]:*Ptger4*[fl/fl] mice injected with vehicle or 10 mg/kg/day SW033291 for 1 month. N ≥ six per group. *p < 0.05, **p<0.01, and N.S. indicates not significant. Statistical significance was determined by two-way analysis of variance.

The online version of this article includes the following source data for figure 4:

**Source data 1.** Raw data of quantification of dynamic bone formation, staining of pCREB, perilipin, osteocalcin, cAMP concentration and gene expression.

Injection of BIBO3304 significantly increased levels of pCREB in osteocalcin[+] osteoblasts at the bone surface in *Avil*[Cre]:*Ptger4*[fl/fl] mice (*Figure 6E*), and cAMP concentration was also increased (*Figure 6F*). Importantly, *Avil*[Cre]:*Ptger4*[fl/fl] mice injected with BIBO3304 had significantly increased expression of osteogenesis markers *Runx2*, *Alp*, and *Col1a1* (*Figure 6G*). FA oxidation is required for osteoblast differentiation (*Kim et al., 2017*). The expression of FA oxidation factors *Fatp1* and *Cpt1b* in osteoblasts decreased in *Avil*[Cre]:*Ptger4*[fl/fl] mice but was rescued with injection of BIBO3304 (*Figure 6H*). Moreover, the levels of FAs were significantly increased in *Avil*[Cre]:*Ptger4*[fl/fl] mice and significantly decreased with injection of BIBO3304 (*Figure 6I*), indicating that BIBO3304 stimulated FA oxidation for osteoblast differentiation. In addition, BIBO3304 significantly increased lipolysis marker *Lipe* and *Pnpla2* expression in *Avil*[Cre]:*Ptger4*[fl/fl] mice (*Figure 6J*) and augmented catabolic metabolism of adipocytes in *Avil*[Cre]:*Ptger4*[fl/fl] mice as shown with an increase of pAMPK (*Figure 6K*). Accordingly, accumulated marrow fat droplets in *Avil*[Cre]:*Ptger4*[fl/fl] mice were significantly lower with injection of BIBO3304 relative to that of *Avil*[Cre]:*Ptger4*[fl/fl] mice with vehicle injection (*Figure 6L and M*). Taken together, our data indicate that ascending PGE2/EP4 skeleton interoceptive signaling induces commitment of MSCs to osteoblast lineages, whereas downregulation of NPY expression in the hypothalamus promotes osteoblast differentiation by increasing FA oxidation in osteoblasts and lipolysis in adipocytes as a parallel descending neuroendocrine interoceptive pathway.

## Y1R promotes osteoblast differentiation and bone formation while β2-adrenergic receptor (β2R) induces lineage commitment of MSCs

Our previous studies demonstrated that the commitment of Leptin receptor[+] (LepR[+]) MSCs regulated by PGE2/EP4 skeleton interoception and activation of β2R rescued bone loss and stimulated osteogenic differentiation in *Avil*[Cre]:*Ptger4*[fl/fl] mice (*Chen et al., 2019*; *Hu et al., 2020*). To investigate the different mechanistic effects of PGE2/EP4 interoception and neuronal-endocrine NPY on osteoblastic bone formation, we crossed *Lepr*[Cre] mice with *Rosa26*[lsl-EYFP] mice to generate *Lepr*[Cre]:*Rosa26*[lsl-EYFP] (Lepr;YFP) mice for lineage-tracing the fate of MSCs. Fate mapping assay showed that YFP[+] osterix[+] preosteoblasts and YFP[+] osteocalcin[+] osteoblasts increased significantly with injection of β2R antagonist, propranolol (Prop). Interestingly, injection of high affinity Y1R agonist [Leu[31], Pro[34]]-NPY did not change the number of YFP[+] osterix[+] preosteoblasts relative to vehicle treatment, but YFP[+] osteocalcin[+] osteoblasts decreased significantly, indicating the effect of NPY on osteoblast differentiation but not the commitment of MSCs. Furthermore, co-injection of propranolol and [Leu[31],Pro[34]]-NPY significantly increased YFP[+]osterix[+] preosteoblasts and still significantly decreased YFP[+] osteoblasts relative to control groups (*Figure 7A* through 7C), indicating that LepR[+] MSC commitment was regulated by β2R through skeletal interoception, whereas osteoblastic differentiation and bone formation were promoted by Y1R through interoception-induced expression of NPY. To validate the in vivo observation, we performed colony-forming units–fibroblast (CFU-F) and CFU-osteoblast (CFU-Ob) assays. CFU-F and CFU-Ob assay showed significant increase with propranolol treatment relative to vehicle. [Leu[31], Pro[34]]-NPY treatment significantly reduced CFU-Ob with no significant changes in CFU-F. As expected, the combination of propranolol and [Leu[31], Pro[34]]-NPY significantly increased CFU-F but decreased CFU-Ob (*Figure 7D* through 7F). Therefore, our data show that PGE2/EP4 ascending interoceptive signaling downregulates sympathetic activity and β2R signaling for the of MSCs. In parallel, PGE2/EP4 ascending interoceptive signaling downregulates expression of NPY in hypothalamus to regulate fat and bone metabolism in facilitation of osteoblast differentiation (*Figure 7G*).

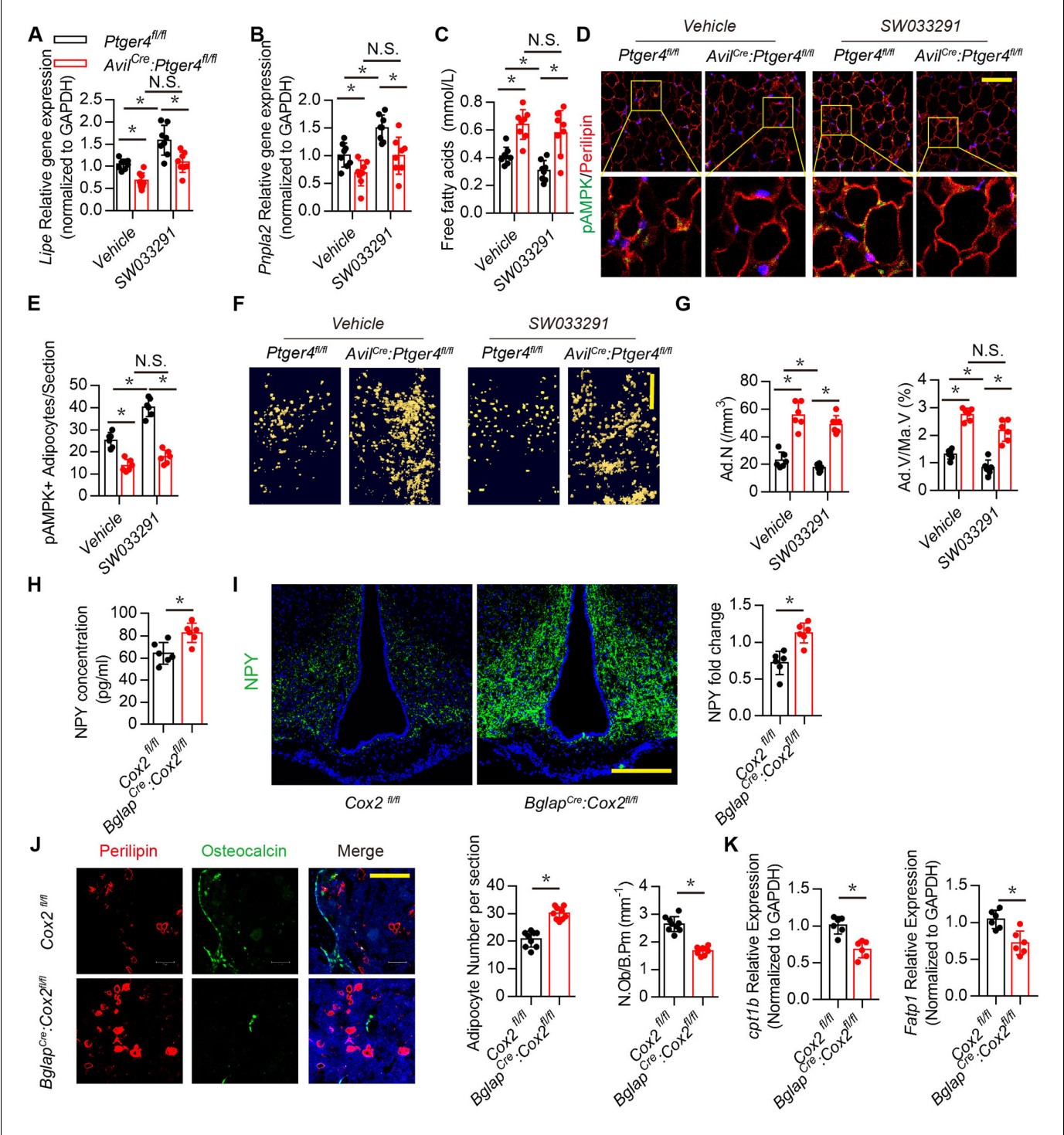

**Figure 5.** Downregulation of NPY by skeletal interoception stimulates lipolysis. RT-PCR quantitative analysis of (**A**) *Lipe* and (**B**) *Pnpla2* expression in the gonadal white adipose tissues from 3-month-old male *Ptger4*^fl/fl^ and *Avil*^Cre^:*Ptger4*^fl/fl^ mice injected with vehicle or 10 mg/kg/day SW033291 for 1 month. (**C**) Quantitative analysis of free FA level in serum from 3-month-old male *Ptger4*^fl/fl^ and *Avil*^Cre^:*Ptger4*^fl/fl^ mice injected with vehicle or 10 mg/kg/d SW033291 for 1 month. (**D**) Representative co-immunofluorescence staining and (**E**) quantitative analysis of pAMPK (*green*) and perilipin (*red*) from the gonadal white adipose tissues of 3-month-old male *Ptger4*^fl/fl^ and *Avil*^Cre^:*Ptger4*^fl/fl^ mice injected with vehicle or 10 mg/kg/day SW033291 for 1 month. (**F**) Representative μCT-detected OsO4-stained images of decalcified femurs and (**G**) quantitative analysis of the number of adipocytes (Ad.N) Ad.V/Ma. V in distal femurs from 3-month-old male *Ptger4*^fl/fl^ and *Avil*^Cre^:*Ptger4*^fl/fl^ mice treated vehicle or 10 mg/kg/day SW033291 for 1 month. Scale bars = 500 μm. (**H**) ELISA analysis of NPY level in serum from 3-month-old *Cox2*^fl/fl^ and *Bglap*^Cre^:*Cox2*^fl/fl^ mice. (**I**) Representative images of immunofluorescence

*Figure 5 continued on next page*

*Figure 5 continued*

staining and quantitative analysis of NPY (*green*) in the ARC of hypothalamus of 3-month-old *Cox2 $^{fl/fl}$* and *Bglap$^{Cre}$:Cox2$^{fl/fl}$* mice. DAPI stains nuclei blue. Scale bars = 50 μm. (J) Representative co-immunofluorescence staining and quantitative analysis (adipocyte number per section and number of osteoblast) of osteocalcin (*green*) and perilipin (*red*) from femurs of 3-month-old male *Cox2 $^{fl/fl}$* and *Bglap$^{Cre}$:Cox2$^{fl/fl}$* mice. Scale bars = 50 μm. (K) RT-PCR quantitative analysis of *Fatp1* and *Cpt1b* expression in femurs from 3-month-old male *Cox2 $^{fl/fl}$* and *Bglap$^{Cre}$:Cox2$^{fl/fl}$* mice N ≥ six per group. *p < 0.05 and N.S. indicates not significant. Statistical significance was determined by the two-way analysis of variance for A-C, E, G.

The online version of this article includes the following source data for figure 5:

**Source data 1.** Raw data of quantification of *Lipe,Pnpla2, cpt1b, Fatp1* gene expression, free fatty acids level, NPY level, pAMPK, NPY, perilipin and osteocalcin staining.

## Discussion

The skeleton, as the largest organ, provides mechanical support for the body and enables locomotion and physical activity. The skeleton is also the reservoir of calcium and minerals as an endocrine organ to regulate calcium and energy metabolism. Interoceptive connections between the brain and peripheral organs are an essential mechanism of CNS control of internal organ activity (*Prescott et al., 2020*; *Umans and Liberles, 2018*). We have reported that osteoblasts-derived PGE2 activates its receptor EP4 in the sensory nerve transmitted through hypothalamic pCREB as ascending interoceptive signaling to tune down sympathetic activity for osteoblast commitment of MSCs (*Chen et al., 2019*). Osteoblastic bone formation is an intensive energy-consuming process, which involves extensive matrix protein synthesis and mineralization in association with angiogenesis and nerve innervation, osteoblasts oxidize fatty acids account as much as 40–80% of the energy yield of glucose consumption (*Adamek et al., 1987*). Here, we found that the PGE2/EP4 ascending interoceptive signaling induces expression of NPY in the hypothalamus to activate lipolysis of adipose tissues to mobilize free FAs for osteoblast differentiation and bone formation as a parallel descending neuroendocrine interoceptive pathway. The hypothalamus regulates whole-body metabolism and energy homeostasis through the sympathetic or neuroendocrine system (*Yadav et al., 2009*; *Confavreux et al., 2009*). While descending sympathetic interoceptive signaling commends osteoblast lineage commitment of MSCs, downregulation of hypothalamic NPY promotes differentiation of the osteoblast committed MSCs for bone formation as neuroendocrine descending interoceptive signaling to activate fat and bone metabolism.

Ascending PGE2/EP4 skeleton interoceptive signaling by injection of SW033291 induced lipolysis and increased bone formation. However, the body weight and lean mass were not changed significantly. Similarly, interruption of ascending interoceptive signaling increased adipose tissue and decreased bone volume in *Avil$^{Cre}$:Ptger4$^{fl/fl}$* mice, and again body weight and lean mass remained unchanged. Metabolism of FAs is required for differentiation of osteoblasts (*Frey et al., 2015*). The specific regulation between bone and adipose tissue indicates that metabolic use of FAs in osteoblasts is essential for osteoblast differentiation and bone formation. The neuroendocrine descending interoceptive signaling downregulates hypothalamic NPY expression to modulate metabolism between fat and bone marrow adipose tissue. Emerging evidence demonstrated that Arc NPY signaling could inhibit sympathetic tone to downregulate uncoupling protein 1(UCP1) expression in brown adipose tissue via tyrosine hydroxylase-containing (TH) neurons (*Shi et al., 2013*; *Shi et al., 2017*). Single-cell RNA sequencing analysis has revealed that heterogeneity of NPY neurons gave rise to various subsets of NPY neuronal populations that are distinguished by the profile of expression of other neurotransmitters (*Chen et al., 2017*). NPY regulates the hematopoietic stem cell microenvironment in the bone marrow (*Park et al., 2015*; *Park et al., 2016*) and induces adipogenesis also through downregulation of the sympathetic tone (*Chao et al., 2011*).

Mechanistically, elevated PGE2 concentration in the bone marrow activates expression of hypothalamic transcriptional co-repressor SMILE of pCREB to downregulate NPY expression in the ARC. It has been reported that SMILE switches the transcriptional activator pCREB to a repressor in suppression of glucogenesis-related genes (*Lee et al., 2018*). SMILE forms a heterodimer with pCREB to bind on *Npy* promoter in repression *Npy* gene expression. Phosphorylation of CREB elevates NPY expression in the brain (*Wand, 2005*; *Pandey et al., 2005*). Physiologically, NPY messenger ribonucleic acid levels at the ARC increase under starvation and are expressed in a circadian pattern, with peak levels shortly before the onset of the dark phase (*Akabayashi et al., 1994*; *Bi et al.,*

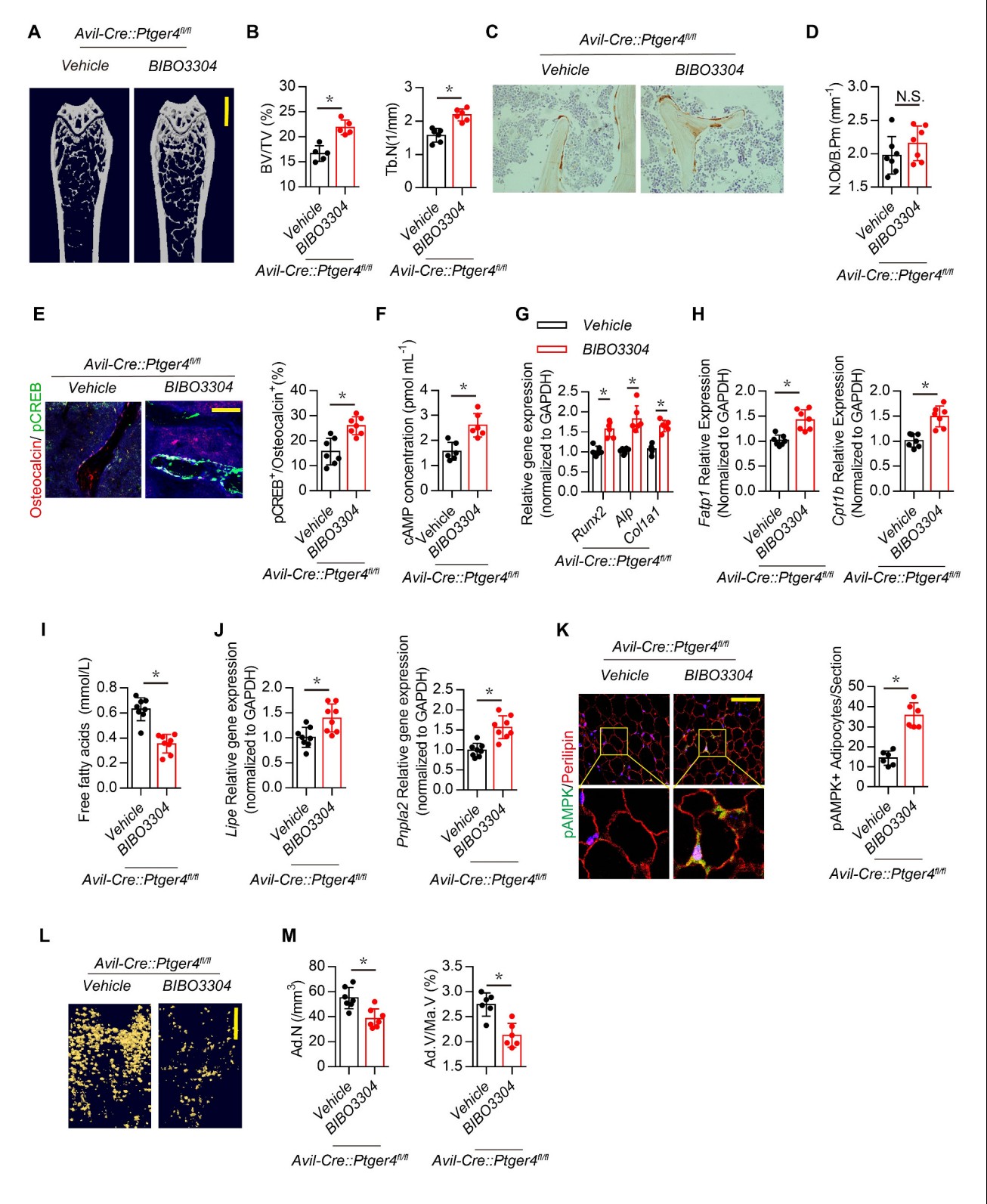

**Figure 6.** Inhibition of Y1R accelerates oxidation of free FAs and rescues bone loss in *Avil^Cre:Ptger4^fl/fl* mice. (**A**) Representative μCT images of femurs from 3-month-old male *Avil^Cre:Ptger4^fl/fl* mice injected with vehicle or 1 mg/kg/day BIBO3304 for 1 month. (**B**) Quantitative analysis of trabecular bone fraction (BV/TV) and trabecular number (Tb.N). Scale bars = 1 mm. (**C**) Representative images of immunostaining and (**D**) quantitative analysis of the osteocalcin^+ cells (*brown*) on trabecular bone surface of femoral bone. (**E**) Representative co-immunofluorescence staining and quantitative analysis of
*Figure 6 continued on next page*

*Figure 6 continued*

pCREB (*green*) and osteocalcin (*red*) from femurs of 3-month-old male *Avil^Cre:Ptger4^fl/fl* mice injected with vehicle or 1 mg/kg/day BIBO3304 for 1 month. Scale bars = 50 μm. (F) Quantitative analysis of cAMP concentration in osteoblasts isolated from 3-month-old male *Avil^Cre:Ptger4^fl/fl* mice injected with vehicle or 1 mg/kg/day BIBO3304 for 1 month. (G, H) RT-PCR analysis of expression of *Runx2, ALlp, Col1a1, Fatp1,* and *Cpt1b* level of bone marrow from 3-month-old male *Avil^Cre:Ptger4^fl/fl* mice injected with vehicle or 1 mg/kg/d BIBO3304 for 1 month. (I) Quantitative analysis of free FA level in serum from 3-month-old male *Avil^Cre:Ptger4^fl/fl* mice injected with vehicle or 1 mg/kg/d BIBO3304 for 1 month. (J) RT-PCR analysis of expression of *Lipe* and *Pnpla2* level of white adipose tissues of 3-month-old male *Avil^Cre:Ptger4^fl/fl* mice injected with vehicle or 1 mg/kg/d BIBO3304 for 1 month. (K) Representative co-immunofluorescence staining and quantitative analysis of pAMPK (*green*) and perilipin (*red*) from gonadal white adipose tissues of 3-month-old male *Avil^Cre:Ptger4^fl/fl* mice injected with vehicle or 1 mg/kg/d BIBO3304 for 1 month. (L) Representative μCT-detected OsO4-stained images of decalcified femurs and (M) quantitative analysis of Ad.N and Ad.V/Ma.V in distal femurs from 3-month-old male *Avil^Cre: Ptger4^fl/fl* mice treated with vehicle or 1 mg/kg/d BIBO3304 for 1 month. Scale bars = 500 μm. N ≥ six per group. *p < 0.05 and N.S. indicates not significant. Statistical significance was determined by Student's t-test.

The online version of this article includes the following source data for figure 6:

**Source data 1.** Raw data of quantification of MicroCT, staining of osteocalcin, pCREB and pAMPK, gene expression, cAMP concentration, free fatty acids level and OsO4 staining.

*2003*; *Ahima et al., 1996*). Indeed, several circulating metabolic hormones, such as insulin and leptin, directly modulate ARC NPY neurons by peripheral signals (*Loh et al., 2017*; *Baskin et al., 1999*). Specifically, the hunger hormone ghrelin increases NPY expression and hence promotes food intake and energy conservation (*Riediger, 2012*; *Tang-Christensen et al., 2004*), whereas leptin, insulin, satiety factor glucagon-like peptide 1, and peptide YY reduce NPY expression to induce satiety and promote energy expenditure. PGE2 also stimulates leptin release (*Fain et al., 2000*). Reduction of NPY in the hypothalamus induced by PGE2/EP4 skeletal interoception shows that oxidation of FAs is essential for osteoblastic bone formation as an intensive energy-consuming process.

The NPY receptor Y1R is expressed widely in the CNS and peripheral nervous system (*Yang et al., 2018*). Global or conditional knockout of Y1R in osteoblasts significantly increases bone volume (*Baldock et al., 2007*). Interestingly, Y1R antagonist BIBO3304 rescued bone loss in *Avil^Cre: Ptger4^fl/fl* mice, which suggests that Y1R signaling is downstream of the NE/β2-adrenergic receptor in PGE2/EP4 interoception. Indeed, skeletal interoception induces commitment of MSCs to osteoblasts, whereas Y1R antagonist BIBO3304 does not change the number of osteocalcin^+ osteoblasts but does increase bone formation by promoting differentiation of committed preosteoblasts. Fate mapping experiments have shown that LepR^+ MSCs-derived preosteoblasts and mature osteoblasts increased significantly with inhibition of the β2-adrenergic receptor. However, Y1R agonist significantly decreased LepR^+ osteocalcin^+ osteoblasts but with no change in LepR^+ preosteoblasts. Moreover, inhibition of β2-adrenergic receptor and activation of Y1R result in an increase of LepR^+ preosteoblasts and a decrease of LepR^+ osteoblasts. Thus, downregulation of hypothalamic NPY induces osteoblast differentiation with the supply of FAs.

## Materials and methods

### Mice and in vivo treatment

The *Rosa26 ^lsl-DTR* mice were purchased from the Jackson Laboratory (007900, Bar Harbor, ME). The *Advillin-Cre* (*Avil^Cre*) mouse strain was kindly provided by Xingzhong Dong (The Johns Hopkins University, Baltimore, MD). The *Ntrk1^fl/fl* mice were obtained from David D. Ginty (Harvard Medical School, Boston, MA). The *Ptger4^fl/fl* mice were obtained from Brian L. Kelsall (National Institutes of Health, Bethesda, MD). Heterozygous male *Avil^Cre* mice (female *Avil^Cre* mice were not used for breeding because of the risk of leakage of TrkA protein into the eggs) were crossed with a *Ntrk1^fl/fl*, *Ptger4^fl/fl*, or *Rosa26 ^lsl-DTR* mouse. The offspring were intercrossed to generate the following genotypes: WT, *Avil^Cre* (Cre recombinase expressed driven by Advillin promoter), *Avil^Cre:Ptger4^fl/fl* (conditional deletion of the EP4 receptor in Advillin cells), *Avil^Cre:Ntrk1^fl/fl*, and *Avil^Cre: Rosa26 ^lsl-DTR*. To generate the inducible sensory denervation mouse model, we injected 8-week-old *Avil^Cre: Rosa26 ^lsl-DTR* mice with 1 μg /kg of DTX three times a week for four consecutive weeks. The *Lepr^Cre:Rosa26^lsl-EYFP* (Lepr;YFP) lineage tracing mice were generated by crossing the *Lepr^Cre* (008320, Jackson lab, Bar Harbor, ME) and *Rosa26^lsl-EYFP* mice (007903, Jackson lab, Bar Harbor, ME). The genotypes of the mice were determined by PCR analyses of genomic DNA, which was extracted from mouse tails

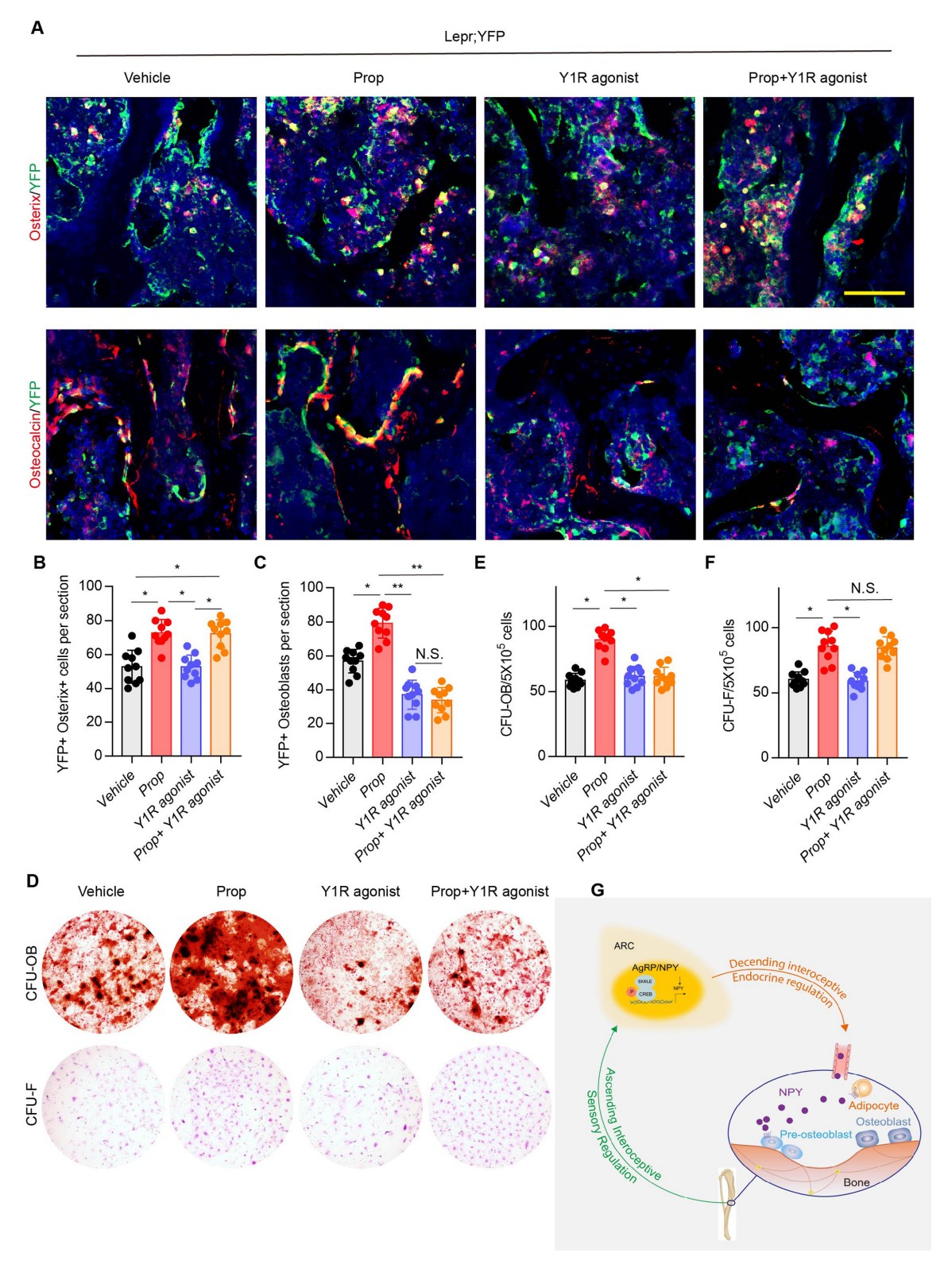

**Figure 7.** β2R regulates MSC commitment and Y1R coordinates osteoblastic formation and mineralization. (**A**) Representative images of immunofluorescence staining of colocalization of Osterix (*red*) and osteocalcin (*red*) with YFP (*green*) (representing LepR+ cells) and (**B, C**) quantitative analysis of (**B**) YFP$^+$ osterix$^+$ cells per section and (**C**) YFP$^+$ osteoblasts per section in femur bone marrow from 3-month-old Lepr;YFP mice treated with propranolol (0.5 mg/kg/day for 6 weeks), 5 μmol/mouse/day [Leu$^{31}$, Pro$^{34}$]-NPY, and a combination of propranolol (0.5 mg/kg/d for 6 weeks) and 5

*Figure 7 continued on next page*

*Figure 7 continued*

umol/mouse/d [Leu$^{31}$, Pro$^{34}$]-NPY for 4 weeks. Scale bars = 50 μm. (**D**) Representative images of alizarin red S–stained CFU-Ob and crystal violet–stained CFU-F. Quantitative analysis of (**E**) CFU-Ob and (**F**) CFU-F MSCs isolated from 3-month-old male Lepr;YFP mice treated with propranolol (0.5 mg/kg/d for 6 weeks), 5 umol/mouse/day [Leu$^{31}$, Pro$^{34}$]-NPY, and a combination of propranolol (0.5 mg/kg/day for 6 weeks) and 5 umol/mouse/d [Leu$^{31}$, Pro$^{34}$]-NPY for 4 weeks. N ≥ six per group. *p < 0.05 and N.S. indicates not significant. Statistical significance was determined by two-way analysis of variance. (**G**) Diagram showing that PGE2/EP4 ascending signal also downregulates expression of neuroendocrine factor NPY, which is secreted into circulation, as the neuroendocrine descending interoceptive signal for bone and fat metabolism.

The online version of this article includes the following source data for figure 7:

**Source data 1.** Raw data of quantification of YFP staining and CFU-OB and CFU-F.

within the primers in the 'Sequence of The Primers' in the supplementary file. All mice were maintained at the animal facility of The Johns Hopkins University School of Medicine (Baltimore, MD). We obtained whole blood samples by cardiac puncture immediately after euthanasia. Serum was collected by centrifuge at 1500 rpm for 15 min and stored at −80℃ before analyses. Mice femurs, brains, dorsal root ganglia, and urine were also collected. Body weight was measured every 3 months.

The drugs and compounds used in this study are as follows: diphtheria toxin (D0564, DTX, Sigma-Aldrich, Saint Louis, MO), SW033291 (S7900, Selleck, Houston, TX), and NPY Y1R inhibitor BIBO3304 (2412, Tocris, Minneapolis, MN). Y1R agonist [Leu$^{31}$, Pro$^{34}$]-NPY (TP2206, Target Mol, Wellesley Hills, MA). Dosages and time courses are noted in the corresponding text and figure legends.

## μCT analyses

Mouse femurs were harvested, and the soft tissue around the bone was removed, followed by fixation overnight using 4% paraformaldehyde. μCT analyses were performed using a high-resolution μCT scanner (1174, SkyScan, Bruker, Kontich, Belgium). The voltage of the scanning procedure was 65 keV with a 153-μA current. The resolution was set to 8.7 μm/pixel. Images were reconstructed using NRecon, version 1.6, software (SkyScan) and analyzed using CTAn, version 1.9, software (SkyScan). We used 3-dimensional model visualization software, CTVol, version 2.0 (SkyScan), to analyze the diaphyseal cortical bone and the metaphyseal trabecular bone parameters of the femurs. We created cross-sectional images of the femur to perform two-dimensional analyses of cortical bone and three-dimensional analyses of trabecular bone. The region of interest of the trabecular bone was defined as beginning from 5% of the femur length proximal to the distal metaphyseal growth plate and extending proximally for another 5% of the total femur length. The trabecular bone volume fraction (BV/TV), trabecular thickness (Tb. Th), trabecular number (Tb. N), and trabecular separation (Tb. Sp) were collected from the three-dimensional analysis data and used to represent the trabecular bone parameters.

## OsO$_4$ staining and μCT analysis

The femurs were harvested from the mice, fixed in 4% phosphate-buffered paraformaldehyde for 48 hr, and decalcified for 2 weeks in 10% ethylenediaminetetraacetic acid (EDTA) at 4℃. The proximal ends of the femurs were cut off and discarded. We incubated the distal part of the femurs in 2% aqueous osmium tetroxide (OsO$_4$, Sigma-Aldrich) for 2 hr in the fume hood. The femurs were rinsed in phosphate buffered saline for 48 hr and then scanned using a high-resolution μCT scanner (1172, Skyscan, Bruker MicroCT) at 6 μm resolution using 45 keV and 177 μA. Quantification of marrow adipose tissue volume, density, and distribution in bone was registered to decalcified bone as previously described (*Hu et al., 2020*).

## Immunohistochemistry and immunofluorescence assay

The femurs were collected and fixed in 4% paraformaldehyde overnight and decalcified using 10% EDTA (pH, 7.4) (0105, Amresco, Dallas, TX) for 21 days. The samples were then dehydrated with 30% sucrose for 24 hr and embedded in paraffin or optimal cutting temperature compound (Sakura Finetek, Torrance, CA). Thick sections were cut as described previously[38]. Briefly, the femurs were fixed for 4 hr with 4% paraformaldehyde at 4℃ and then decalcified at 4℃ using 0.5 M EDTA (pH, 7.4) for 24 hr with constant shaking. The samples were dehydrated in 20% sucrose and 2%

polyvinylpyrrolidone solution for 24 hr and embedded in 8% gelatin (G1890, Sigma-Aldrich) in the presence of 20% sucrose and 2% polyvinylpyrrolidone. Forty μm–thick coronal-oriented sections of the femurs were obtained. For brain section preparation, the whole brain was collected from euthanized mice and fixed with 4% paraformaldehyde for 30 min. Then, the tissue was dehydrated with 20% sucrose for 24 hr, followed by 30% sucrose for 24 hr and sectioned.

Immunostaining was performed using standard protocol. Briefly, the sections were incubated with primary antibodies to mouse Osterix (1:600, ab22552, Abcam, Cambridge, UK), osteocalcin (1:200, M173, Takara Bio, Mountain View, CA), perilipin (1:100, p1873, Sigma-Aldrich), CGRP (1:100, ab81887, Abcam), NPY (1:400, 11976 s, Cell Signaling Technology, Danvers, MA), pCREB (1:100, ab32096, Abcam), pAMPK (2535 s, Cell Signaling Technology), Crebzf (1:50, c111755, Assay Biotech, Sunnyvale, CA), and GFP (1:800, ab13970, Abcam) overnight at 4℃. A horseradish peroxidase–streptavidin detection kit (Dako, Agilent, Santa Clara, CA) was used in immunohistochemical procedures to detect immuno-activity, followed by counterstaining with hematoxylin (S3309, Dako). Fluorescence-conjugated secondary antibodies were used in immunofluorescent procedures to detect fluorescent signals after counterstaining with DAPI (H-1200, Vector, Burlingame, CA). We used a LSM 780 confocal microscope (Zeiss, Oberkochen, Germany) or an Olympus BX51 microscope (Olympus, Tokyo, Japan) for sample image capturing. Quantitative histomorphometric analysis was performed by using OsteoMeasure XP software (OsteoMetric, Decatur, GA) in a blinded fashion.

A double-labeling procedure was performed to measure dynamic bone formation. Briefly, we injected 0.1% calcein (C0875, Sigma-Aldrich) in phosphate buffered saline at a concentration of 10 mg/kg into the mice subcutaneously 7 days and 1 day before sacrifice. The double-labeling images of undecalcified bone slices were captured using a fluorescence microscope. We analyzed trabecular bone formation in four randomly selected visual fields in the distal metaphyseal area of the femur.

## ChIP and antibodies

ChIP was performed according to instructions from the Pierce Agarose ChIP Kit (26156, Thermo Fisher Scientific, Waltham, MA) with ChIP-grade antibody SMILE (9198, Cell Signaling Technology). Briefly, we added cells with formaldehyde to cross-link proteins to DNA, and the cells were lysed in 1.5 mL lysis buffer (50 mM HEPES, pH 7.5, 140 mM NaCl; 1 mM EDTA; 1% Triton X-100; 0.1% sodium deoxy cholate; 0.1% sodium dodecyl sulfate). Cell lysates were sonicated at 2 s on/15 s off for three rounds using a Bioruptor ultrasonic cell disruptor (Diagenode, Denville, NJ) to shear genomic DNA to a mean fragment size of 150–250 bp. Of the sample, 1% was removed for use as input control. ChIP was performed according to the protocol provided by the Simple Chip Enzymatic Chromatin IP Kit (Cell Signaling Technology) using antibodies to pCREB (Cell Signaling Technology). Anti-RNA polymerase II and control IgG were used as positive and negative controls, respectively. After washing and de-crosslinking, the precipitated DNA was purified using a QIA quick PCR purification kit (Qiagen, Hilden, Germany).

## ChIP-quantitative PCR

ChIP-quantitative PCR (qPCR) was performed using SYBR green PCR Master Mix and 7900 HT Fast Real-Time PCR System (Applied Biosystems, Foster City, CA). Primers for Frag 1, 2, and 3 of periostin were used (see *Table 1* for primer sequences). Absolute quantification was performed, and enrichment was expressed as a fraction of the whole-cell extract control.

## Quantitative real-time polymerase chain reaction (qPCR)

Total RNA was purified from cells in culture or tissues using TRIzol (15596026, Invitrogen, Carlsbad, CA) following the manufacturer's protocol. We performed qPCR using the Taq SYBR Green Power PCR Master Mix (A25777, Invitrogen) on a CFX Connect instrument (Bio-Rad Laboratories, Hercules, CA); *Gapdh* amplification was used as an internal control. Dissociation curve analysis was performed for every experiment. Sequences of the primers used for each gene are available in the 'Sequence of The Primers' in the supplementary file.

## Statistical analysis

All data analyses were performed using SPSS, version 15.0, software (IBM Corp., Armonk, NY). Data are presented as means ± standard errors of the mean. For comparisons between two groups, we used two-tailed Student $t$-tests. For comparisons among multiple groups, we used two-way analysis of variance. All relevant data are available from the authors.

## Study approval

All animal experiments were performed following NIH policies on the use of laboratory animals. All experimental protocols were approved by the Animal Care and Use Committee of The Johns Hopkins University.

## Acknowledgements

This research was supported by the National Institute on Aging of the National Institutes of Health under Award Number P01AG066603 and 1R01 AG068997 (to X C). For their editorial assistance, we thank Jenni Weems, MS, Kerry Kennedy, BA, and Rachel Box, MS, in the Editorial Services group of The Johns Hopkins Department of Orthopaedic Surgery.

## Additional information

### Funding

| Funder | Grant reference number | Author |
|---|---|---|
| National Institute on Aging | P01AG066603 | Xu Cao |
| National Institute on Aging | 1R01 AG068997 | Xu Cao |

The funders had no role in study design, data collection and interpretation, or the decision to submit the work for publication.

### Author contributions

Xiao Lv, Data curation, Investigation, Methodology, Project administration; Feng Gao, Data curation, Formal analysis; Tuo Peter Li, Data curation; Peng Xue, Investigation, Methodology; Xiao Wang, Methodology, Project administration; Mei Wan, Validation, Visualization; Bo Hu, Resources, Methodology; Hao Chen, Resources, Writing - original draft; Amit Jain, Visualization, Writing - review and editing; Zengwu Shao, Project administration, Writing - review and editing; Xu Cao, Conceptualization, Supervision

### Author ORCIDs

Xiao Lv 
Tuo Peter Li 
Xiao Wang 
Mei Wan 
Xu Cao 

### Ethics

Animal experimentation: This study was performed in strict accordance with the recommendations in the Guide for the Care and Use of Laboratory Animals of the National Institutes of Health. All of the animals were handled according to approved institutional animal care and use committee (IACUC) protocols (MO18M298) of the Johns Hopkins University.

### Decision letter and Author response

Decision letter https://doi.org/10.7554/eLife.70324.sa1
Author response https://doi.org/10.7554/eLife.70324.sa2

## Additional files

**Supplementary files**

- Supplementary file 1. Sequence of primer.
- Transparent reporting form

### Data availability

All data generated or analysed during this study are included in the manuscript and supporting files. Source data files have been provided for Figures 1–7.

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
