## [Decision Letter]

**Acceptance summary:**

This paper, which documents clearly the dual role of hypothalamic NYP in the regulation of bone and fat metabolism, is conceptually novel and is a valuable extension to the evolving concept of how the skeleton is connected with the brain. Potential therapeutic implications may emerge from this study.

---

## [Author Response]

[Editors' note: we include below the reviews that the authors received from another journal, along with the authors’ responses.]

We would like to thank the reviewers for their thoughtful and constructive comments for our manuscript. We have addressed all the questions and comments brought forth through additional experimentation and clarification.

Reviewer A:This is an interesting study with a sophisticated methodological approach, carefully conducted studies, and a large body of insightful results. Several points in the discussion are overstated and not supported by actual data. Some areas of the manuscript are confusing to read, in part because of the imprecise language.Major comments:1. As most other Cre promoters, Advillin^Cre^ is not specific for sensory neurons but found in a variety of different neuronal populations (see Hunter et al. 2018). The authors should provide evidence that EP4 expression was not affected in neuronal populations other than sensory neurons. In this respect, parasympathetic/sympathetic neurons might be of particular interest.

Thanks for your suggestion. We understand that Advilin is also expressed in the sympathetic and parasympathetic nerves. In this experiment, *Avil^Cre^:Ptger4^fl/fl^* were used to examine ascending PGE2 signaling in sensory nerve through EP4 receptor to the hypothalamus. Sympathetic and parasympathetic nerves mainly secrete neurotransmitters as descending signaling. Advilin^Cre^ has been used for labelling sensory nerve in similar situation of different studies (1-5).

Reference:

(1) Choi S, Hachisuka J, Brett MA, Magee AR, Omori Y, Iqbal NU, Zhang D, DeLisle MM, Wolfson RL, Bai L, Santiago C, Gong S, Goulding M, Heintz N, Koerber HR, Ross SE, Ginty DD. Parallel ascending spinal pathways for affective touch and pain. Nature. 2020 Nov;587(7833):258-263.

(2) Neubarth NL, Emanuel AJ, Liu Y, Springel MW, Handler A, Zhang Q, Lehnert BP, Guo C, Orefice LL, Abdelaziz A, DeLisle MM, Iskols M, Rhyins J, Kim SJ, Cattel SJ, Regehr W, Harvey CD, Drugowitsch J, Ginty DD. Meissner corpuscles and their spatially intermingled afferents underlie gentle touch perception. Science. 2020 Jun 19;368(6497):eabb2751.

(3) Chuang YC, Lee CH, Sun WH, Chen CC. Involvement of advillin in somatosensory neuron subtype-specific axon regeneration and neuropathic pain. Proc Natl Acad Sci U S A. 2018 Sep 4;115(36):E8557-E8566.

(4) Zhou X, Wang L, Hasegawa H, Amin P, Han BX, Kaneko S, He Y, Wang F. Deletion of PIK3C3/Vps34 in sensory neurons causes rapid neurodegeneration by disrupting the endosomal but not the autophagic pathway. Proc Natl Acad Sci U S A. 2010 May 18;107(20):9424-9.

(5) Hasegawa H, Abbott S, Han BX, Qi Y, Wang F. Analyzing somatosensory axon projections with the sensory neuron-specific Advillin gene. J Neurosci. 2007 Dec 26;27(52):14404-14.

2. The authors did not causally prove that an "interoception" axis between brain and bone actually exists. They show that injection of an 15PGDH inhibitor elicits accumulation of PGE2 in the bone marrow, but not in the hypothalamus. Since the substance was delivered systemically, PGE2 could potentially accumulate in any other organ of the body to elicit relevant changes via EP4. Furthermore, how do the authors know that PGE2 measured in the bone marrow is actually derived from bone cells rather than leukocyte precursors or similar? From this perspective, I also do not understand why the knock-out of EP4 in sensory neurons would strengthen the authors' "skeleton" hypothesis since prostaglandins activating the receptor could theoretically stem from any other tissue in the body rather than bone. To causally prove the authors' idea, one would have to knock-out 15PGDH specifically in bone cells/osteoblasts. If the authors' hypothesis was correct, then the 15PGDH inhibitor effects should be lost in mice with osteoblast-specific knock out of the enzyme.

Thanks for your questions. Demonstration of “interoception" axis between brain and bone have been published in our previous publication (6, 7).

Specifically, we have shown that PGE2 derived from osteoblasts is primarily involved in sensory nerve regulation of bone formation (6). In the revised manuscript, we have introduced *Bglap^Cre^:Cox2^fl/fl^*, which specifically knockout COX2 in osteoblasts (Figure5 H-K). *Bglap^Cre^:Cox2^fl/fl^* also showed significantly increased NPY expression in ARC companied by bone loss and adipocytes accumulation, Importantly, our previous study found that the effect of SW033291 on osteogenesis induction and adipogenesis inhibition were abrogated in *Bglap^Cre^:Cox2^fl/fl^* mice (7). Therefore, PGE2 primarily secreted by osteoblasts for bone and fat metabolism regulation by PGE2-EP4 skeleton interoception.

To determine the potential PGE2 origin besides osteoblasts within bone marrow, we also examined the *DMP1^Cre^:Cox2^fl/fl^* mice which specifically knock COX2 in osteocytes, *DMP1^Cre^:Cox2^fl/fl^* eliminated PGE2 secretion by osteocytes. We found that NPY expression in the ARC, bone and fat phenotype were unchanged in the *DMP1^Cre^:*Cox2^fl/fl^ mice relative to the control group (Figure 2—figure supplement 1A-D). Osteocytes derived PGE2 did not contribute to the sensory nerve EP4 skeleton interoception in regulate bone and fat metabolism through hypothalamic NPY. COX2 mainly expressed on the bone surface, which colocalized with the osteoblasts (6). Take together, these results reveal that osteoblast derived PGE2 regulate NPY expression in the ARC and bone-fat metabolism through EP4 receptor on sensory nerve.

Reference:

(6) Chen H, Hu B, Lv X, Zhu S, Zhen G, Wan M, Jain A, Gao B, Chai Y, Yang M, Wang X, Deng R, Wang L, Cao Y, Ni S, Liu S, Yuan W, Chen H, Dong X, Guan Y, Yang H, Cao X. Prostaglandin E2 mediates sensory nerve regulation of bone homeostasis. Nat Commun. 2019 Jan 14;10(1):181.

(7) Hu B, Lv X, Chen H, Xue P, Gao B, Wang X, Zhen G, Crane JL, Pan D, Liu S, Ni S, Wu P, Su W, Liu X, Ling Z, Yang M, Deng R, Li Y, Wang L, Zhang Y, Wan M, Shao Z, Chen H, Yuan W, Cao X. Sensory nerves regulate mesenchymal stromal cell lineage commitment by tuning sympathetic tones. J Clin Invest. 2020 Jul 1;130(7):3483-3498.

3. Pertinent to point 2, the authors show that 15PGDH inhibitor administration increases hypothalamic pCREB abundance. Again, no compelling evidence is provided supporting the idea that this effect arises from bone-specific production of PGE2.

Thanks for your questions, to address these concerns, we detected pCREB expression in ARC in the *Bglap^Cre^:*Cox2^fl/fl^ and relative WT mice. We found pCREB expression decreased significantly in the *Bglap^Cre^:*Cox2^fl/fl^ relative to the WT group (Figure2—figure supplement 1E). Therefore, PGE2 derived from osteoblasts specifically for the CREB phosphorylation in ARC in hypothalamus.

4. This group has published other aspects of this story in two previous papers in Nature Communications (2019) and JCI (2020). While there is no duplication in publishing, it raises the issue of novelty.

In this study, we investigated the ascending skeleton interoception regulation of hypothalamic NPY. Hypothalamic NPY regulated by PGE2/EP4 skeleton ascending interoceptive signaling induces lipolysis for osteoblastic bone formation.

5. Language editing is warranted. While the Acknowledgements mention that this manuscript has been sent to the 'Editorial Services' of the Department of Orthopedic Surgery at Johns Hopkins, the writing is still substandard and the conclusions of the paper are hard to understand.

In the revised version, we carefully edited the manuscript.

Minor comments:1. Include validation of successful gene knock-out in main figure.

The genetic knock-out efficiency experiment has been moved into the main figure.

2. Discrepancy between the magnitude of effect in IF pictures shown in figure 1A and corresponding quantification; possibly adapt.

We have checked the magnitude and scale bar in the figure 1, each panel were correct and consistent.

3. Why do the authors suggest that injecting a herpes virus in the femur paralleled by its accumulation in the brain proves a physiologically relevant "interoception" axis between bone and brain? Extrapolating virus-associated alterations as evidence for physiological regulations appears speculative.

Virus-associated trace experiment including anterograde trace and retrograde trace, which was recognized as a classical and reliable evidence for the neurocircuit between different nucleis in the brain and neurocircuit between peripheral organs and the central nervous system (8-10). Importantly, there were no virus-associated trace experiment have ever done before for skeleton-brain route. Here, we firstly recruited the anterograde multisynapitc tracer herpes virus injected in the femur bone marrow in mice, then detected the infected neurons at the hypothalamus area to build up the anatomical foundation of the skeleton-hypothalamus interoception route.

Reference:

(8) Han W, Tellez LA, Perkins MH, Perez IO, Qu T, Ferreira J, Ferreira TL, Quinn D, Liu ZW, Gao XB, Kaelberer MM, Bohórquez DV, Shammah-Lagnado SJ, de Lartigue G, de Araujo IE. A Neural Circuit for Gut-Induced Reward. Cell. 2018 Oct 18;175(3):887-888.

(9) Chen S, He L, Huang AJY, Boehringer R, Robert V, Wintzer ME, Polygalov D, Weitemier AZ, Tao Y, Gu M, Middleton SJ, Namiki K, Hama H, Therreau L, Chevaleyre V, Hioki H, Miyawaki A, Piskorowski RA, McHugh TJ. A hypothalamic novelty signal modulates hippocampal memory. Nature. 2020 Oct;586(7828):270-274.

(10) Bai L, Mesgarzadeh S, Ramesh KS, Huey EL, Liu Y, Gray LA, Aitken TJ, Chen Y, Beutler LR, Ahn JS, Madisen L, Zeng H, Krasnow MA, Knight ZA. Genetic Identification of Vagal Sensory Neurons That Control Feeding. Cell. 2019 Nov 14;179(5):1129-1143.

4. How do the authors explain the discrepancy between increase in fat mass in the presence of unaltered lean mass and body weight? If fat mass increases, while lean mass remains stable, one would expect an increase in body weight.

We found that fat mass increased companied by bone loss in the *Avil^Cre^:Ptger4^fl/fl^* relative to the WT mice. The bone loss was compensated with the fat mass increase, resulting in unchanged body weight.

5. Statements such as 'During the past 15 years, hallmark endocrine hormones such as insulin and parathyroid hormone have been recognized as master regulators of energy metabolism.' are awkward and wrong.

We have changed the statement into “During the past 15 years, endocrine hormones, such as insulin and parathyroid hormone, have been recognized as master regulators of energy metabolism”.

Reviewer B:This manuscript seeks to determine whether PGE2/EP4 ascending interoceptive signaling from the bone to the CNS contributes to the regulation of bone and adipose metabolism. Mice with floxed TrkA were crossed with sensory neuron-specific cre mice (Advilin-Cre) to generate mice with sensory denervation. This denervation would be profound as indicated by staining showing the dorsal root ganglia of these mice. The main limitation of this study is that it seemingly concludes that the important afferent signaling is coming from the bone without providing any specific test of this hypothesis. For example, it is stated that, "To examine whether NPY expression in the hypothalamus is regulated by skeletal interoception, we generated sensory nerve EP4 knockout mice (generated using advilin-cre mice). This would knockout EP4 in all sensory neurons not just those arising from bone. If this is specific to bone then this must be demonstrated. Also, the use of peripheral SW033291 should elevate PGE2 throughout the body not just in bone marrow.The use of the terminology "skeleton interoceptive signaling" implies afferent input. The use of the terminology of "descending PGE2/EP4 skeleton interoceptive signaling" is confusing.

The afferent signaling from bone to CNS interoception have been demonstrated and published in our previous publication (11, 12).

Specifically, we have shown that PGE2 derived from osteoblasts is primarily involved in sensory nerve regulation of bone formation (4). In the revised manuscript, we have introduced *Bglap^Cre^:Cox2^fl/fl^*, which specifically knockout COX2 in osteoblasts. *Bglap^Cre^:Cox2^fl/fl^* also showed significantly increased NPY expression in ARC companied by bone loss and adipocytes accumulation (Figure 5 H-K), Importantly, our previous study found that the effect of SW033291 on osteogenesis induction and adipogenesis inhibition were abrogated in *Bglap^Cre^:Cox2^fl/fl^* mice (12). Therefore, PGE2 primarily secreted by osteoblasts for bone and fat metabolism regulation by PGE2-EP4 skeleton interoception.

To determine the potential PGE2 origin besides osteoblasts within bone marrow, we also examined the *DMP1^Cre^:Cox2^fl/fl^* mice which specifically knock COX2 in osteocytes, *DMP1^Cre^:Cox2^fl/fl^* eliminated PGE2 secretion by osteocytes. We found that NPY expression in the ARC, bone and fat phenotype were unchanged in the *DMP1^Cre^:Cox2^fl/fl^* mice relative to the control group (Figure 2—figure supplement 1A-D). These results showed that osteocytes derived PGE2 did not contribute to the sensory nerve EP4 skeleton interoception in regulate bone and fat metabolism through hypothalamic NPY. Also, according to our previous work (11), We found that COX2 positive cells mainly expressed on the bone surface, which colocalized with the osteoblasts. Taken together, these results indicated that osteoblast derived PGE2 regulate NPY expression in the ARC and bone-fat metabolism through EP4 receptor on sensory nerve.

The use of the terminology "ascending interoceptive signaling" and “descending interoceptive signaling” were suggested in recent review article (13). In the revised manuscript, the terminology was carefully evaluated and revised.

Reference:

(11) Chen H, Hu B, Lv X, Zhu S, Zhen G, Wan M, Jain A, Gao B, Chai Y, Yang M, Wang X, Deng R, Wang L, Cao Y, Ni S, Liu S, Yuan W, Chen H, Dong X, Guan Y, Yang H, Cao X. Prostaglandin E2 mediates sensory nerve regulation of bone homeostasis. Nat Commun. 2019 Jan 14;10(1):181.

(12) Hu B, Lv X, Chen H, Xue P, Gao B, Wang X, Zhen G, Crane JL, Pan D, Liu S, Ni S, Wu P, Su W, Liu X, Ling Z, Yang M, Deng R, Li Y, Wang L, Zhang Y, Wan M, Shao Z, Chen H, Yuan W, Cao X. Sensory nerves regulate mesenchymal stromal cell lineage commitment by tuning sympathetic tones. J Clin Invest. 2020 Jul 1;130(7):3483-3498.

(13) Chen WG, Schloesser D, Arensdorf AM, Simmons JM, Cui C, Valentino R, Gnadt JW, Nielsen L, Hillaire-Clarke CS, Spruance V, Horowitz TS, Vallejo YF, Langevin HM. The Emerging Science of Interoception: Sensing, Integrating, Interpreting, and Regulating Signals within the Self. Trends Neurosci. 2021 Jan;44(1):3-16. doi: 10.1016/j.tins.2020.10.007.

Reviewer C:Endocrine regulation of bone metabolism has been very well studied, but not much is known about the neural regulation of bone. Clinical and basic research evidence suggest that nerves are likely important in controlling bone metabolism upstream of endocrine regulators. In this manuscript, the authors find that sensory nerve sensing of PGE2 concentrations in bone activates EP4 signaling. Specifically, PGE2/EP4 signaling in the sensory circuit induces phosphorylation of CREB in hypothalamus. NPY downregulation in the ARC regulates bone and fat metabolism, as a neuroendocrine interoceptive signal. Overall, this is highly innovative study, and the findings are clinically very relevant. The experiments were well designed, and results are convincing. I have the following comments that should help improve the manuscript.1. NPY regulates both bone and fat metabolism. Particularly, free fatty acids are directly utilized for osteoblast differentiation. This could be a good reason for the regulation of fat and bone by NPY. However, it is clear why osteoblasts utilize fat and how NPY coordination of the osteoblast activities are important. This reasoning needs further elaboration.

Thanks for the suggestions. Firstly, osteoblasts consume free fatty acid account as much as 40-80% of the energy yield glucose consumption (14). Importantly, free fatty acid oxidation is required for osteoblast differentiation (15). Inhibition of NPYY1R in the osteoblast promote osteoblastic differentiation (16). In our study, we found that free fatty acid related gene and lipolysis related gene were significantly increased after administrating with NPYY1R inhibitor. These results showed that inhibition of NPYY1R promote osteoblast differentiation by increasing fatty acid oxidation in the osteoblasts and lipolysis in adipocytes.

Reference:

(14) Adamek G, Felix R, Guenther HL, and Fleisch H. Fatty acid oxidation in bone tissue and bone cells in culture. Characterization and hormonal influences. The Biochemical journal. 1987;248(1):129-37.

(15) Frey JL, Li Z, Ellis JM, Zhang Q, Farber CR, Aja S, et al. Wnt-Lrp5 signaling regulates fatty acid metabolism in the osteoblast. Molecular and cellular biology. 2015;35(11):1979-91.

(16) Sousa DM, Martins PS, Leitao L, Alves CJ, Gomez-Lazaro M, Neto E, et al. The lack of neuropeptide Y-Y1 receptor signaling modulates the chemical and mechanical properties of bone matrix. FASEB journal. 2020;34(3):4163-77.

2. The PGE2 degrading enzyme SW033291 was used to elevate PGE2 concentration. Injection of SW033291 would increase the level of PGE2 in different tissue. It is unclear how bone specificity is achieved.

It is true that SW033291 increased the PGE2 level in different tissue. Therefore, we introduce we have introduced *Bglap^Cre^:Cox2^fl/fl^*, which specifically knockout COX2 in osteoblasts. *Bglap^Cre^:Cox2^fl/fl^*, also showed significantly increased NPY expression in ARC companied by bone loss and adipocytes accumulation (Figure 5H-K), Importantly, our previous study found that the effect of SW033291 on osteogenesis induction and adipogenesis inhibition were abrogated in *Bglap^Cre^:Cox2^fl/fl^*, mice (12). These results demonstrated that PGE2 primarily secreted by osteoblasts for bone and fat metabolism regulation by PGE2-EP4 skeleton interoception signal. To determine the potential PGE2 origin besides osteoblasts within bone marrow, we also examined the *DMP1^Cre^:Cox2^fl/fl^* mice which specifically knock COX2 in osteocytes, *DMP1^Cre^:Cox2^fl/fl^* eliminated PGE2 secretion by osteocytes. We found that NPY expression in the ARC, bone and fat phenotype were unchanged in the *DMP1^Cre^:Cox2^fl/fl^* mice relative to the control group. (Figure 2—figure supplement 1A-D)

3. In Figure 3 the authors show that phosphorylation of CREB facilitates binding to the repressor SMILE to inhibit NPY expression. Does SMILE also bind directly to the NPY promoter?

As the corepressor, SMILE do not directly bind to the NPY promoter, SMILE only bind to the NPY promoter in the form of the heterodimer with pCREB (17,18).

Reference:

(17) Lee JM, Han HS, Jung YS, Harris RA, Koo SH, and Choi HS. The SMILE transcriptional corepressor inhibits cAMP response element-binding protein (CREB)-mediated transactivation of gluconeogenic genes. The Journal of biological chemistry. 2018;293(34):13125-33.

(18) Misra J, Chanda D, Kim DK, Li T, Koo SH, Back SH, et al. Curcumin differentially regulates endoplasmic reticulum stress through transcriptional corepressor SMILE (small heterodimer partner-interacting leucine zipper protein)-mediated inhibition of CREBH (cAMP responsive element-binding protein H). The Journal of biological chemistry. 2011;286(49):41972-84.

4. In Figure 4C, the co-staining pCREB and osteocalcin is not clear. Better images are required.

As suggested, we have changed the image.

5. BIB03304 stimulated bone volume, but osteoblast number was increased. Osteoclast numbers must be examined in Figure 6C.

Thanks for the suggestion, we have added the trap positive osteoclast staining.

6. pCREB containing with osteocalcin is not very clear in Figure 6E.

Thanks for the suggestion, we have changed the image.

7. There are numerous grammatic and typos in the manuscript.

In the revised manuscript, we have carefully edited the manuscript.